# Convergence of Policy Mirror Descent
# Beyond Compatible Function Approximation

Uri Sherman [1]   Tomer Koren [1][2]   Yishay Mansour [1][2]

## Abstract

Modern policy optimization methods roughly follow the policy mirror descent (PMD) algorithmic template, for which there are by now numerous theoretical convergence results. However, most of these either target tabular environments, or can be applied effectively only when the class of policies being optimized over satisfies strong closure conditions, which is typically not the case when working with parametric policy classes in large-scale environments. In this work, we develop a theoretical framework for PMD for general policy classes where we replace the closure conditions with a strictly weaker variational gradient dominance assumption, and obtain upper bounds on the rate of convergence to the best-in-class policy. Our main result leverages a novel notion of smoothness with respect to a local norm induced by the occupancy measure of the current policy, and casts PMD as a particular instance of smooth non-convex optimization in non-Euclidean space.

## 1. Introduction

Modern policy optimization algorithms (Peters & Schaal, 2006; 2008; Lillicrap, 2015; Schulman et al., 2015; 2017) operate by solving a sequence of stochastic optimization problems, each of which being roughly equivalent to:

$$\pi^{k+1} \leftarrow \arg\min_{\pi \in \Pi} \mathbb{E}_{s \sim \mu^k} \left[ \left\langle \widehat{Q}_s^k, \pi_s \right\rangle + \frac{1}{\eta} B(\pi_s, \pi_s^k) \right], \quad (1)$$

where $\mu^k$ is a state probability measure (typically related, or equal to, the occupancy measure of the current policy $\pi^k$) from which sampling is granted through interaction with the environment; $\widehat{Q}^k$ is an estimate of the action-value

function of $\pi^k$, and $B$ is a distance-like function employed to regularize the update so as to not stray too far from $\pi^k$. The solution to Equation (1) is usually produced by optimizing a parametric neural network model $\pi_\theta$ (known as the actor, or policy network) via multiple steps of stochastic gradient descent, and consequently, the policy class $\Pi$ is the set of policies representable by the model; $\Pi = \{\pi_\theta \mid \theta \in \mathbb{R}^p\}$, where $p$ denotes the number of parameters in the network.

Contemporary theoretical analyses of this algorithm (Shani et al., 2020; Agarwal et al., 2021; Xiao, 2022; Ju & Lan, 2022; Zhan et al., 2023; Yuan et al., 2023; Alfano et al., 2023) all have their roots in the online Markov decision process (MDP) framework, and roughly build on decomposing Equation (1) state-wise and casting the problem as a collection of independent online mirror descent steps (Even-Dar et al., 2009). The disadvantage of such an approach lies in the requirement that the update step be exact (or almost exact) in each state independently, effectively limiting the applicability of such analyses to policy classes that are *complete*, (i.e., $\Pi = \Delta(\mathcal{A})^{\mathcal{S}}$), or otherwise satisfy strong closure conditions.

Largely, papers that develop convergence upper bounds for algorithms following Equation (1), commonly known as Policy Mirror Descent (PMD; Tomar et al., 2020; Xiao, 2022; Lan, 2023), fall into two main categories. The first category includes studies that target the tabular setup (e.g., Geist et al., 2019; Shani et al., 2020; Agarwal et al., 2021; Xiao, 2022; Johnson et al., 2023; Lan, 2023; Zhan et al., 2023), where no sampling distribution $\mu^k$ is involved (or it has no effect) and updates are performed in a per-state manner. The second category consists of papers that consider parametric policy classes (e.g., Agarwal et al., 2021; Alfano & Rebeschini, 2022; Ju & Lan, 2022; Yuan et al., 2023; Alfano et al., 2023; Xiong et al., 2024) often building—either directly or indirectly—on the compatible function approximation framework (Sutton et al., 1999). As such, these works essentially assume that the update in Equation (1) remains "close" to the one that would have been performed over the complete policy class (see Section 1.2 for further discussion). This state of affairs is (at least partially) due to the fact that policy gradient methods in the general policy class setting are prone to local optima (Bhandari & Russo,

[1]Blavatnik School of Computer Science and AI, Tel Aviv University, Tel Aviv, Israel [2]Google Research, Tel Aviv, Israel. Correspondence to: Uri Sherman <urisherman@mail.tau.ac.il>.

*Proceedings of the 42nd International Conference on Machine Learning*, Vancouver, Canada. PMLR 267, 2025. Copyright 2025 by the author(s).

2024), and as a result, structural assumptions are necessary to establish global optimality guarantees.

The present paper aims to establish *best-in-class* convergence of PMD (Equation (1)) for general policy classes, relaxing the stringent closure conditions and assuming instead *variational gradient dominance* (VGD) condition (Bhandari & Russo, 2024; Agarwal et al., 2021; Xiao, 2022). It can be shown that a general form of closure conditions implies VGD and that the converse does not hold, hence it is a strict relaxation of the setup assumptions (see detailed discussion in Section 1.2 and Appendix A). Our main result features a novel analysis technique that casts Equation (1) as a particular instance of smooth non-convex optimization in a non-Euclidean space, where the smoothness of the objective is w.r.t. a *local* norm induced by the current policy occupancy measure. Importantly, this approach leads to rates independent of the cardinality of the state space. In contrast, previous results that establish convergence of gradient based methods (though not of PMD; e.g., Agarwal et al., 2021; Bhandari & Russo, 2024; Xiao, 2022) that are applicable in our setting, lead to bounds that depend on the size of the state-space, thus rendering them useful only in tabular setups.

## 1.1. Main results

We consider the problem of finding an (approximately) optimal policy in a discounted MDP $\mathcal{M} = (\mathcal{S}, \mathcal{A}, \mathbb{P}, r, \gamma, \rho_0)$ within a general policy class $\Pi \subset \Delta(\mathcal{A})^{\mathcal{S}}$. We assume the action set is finite $A := |\mathcal{A}|$, and denote the effective horizon by $H := \frac{1}{1-\gamma}$. Our goal is to minimize the value $V(\pi)$, defined as the long term discounted cost (we interpret $r: \mathcal{S} \times \mathcal{A} \to [0, 1]$ as measuring regret, or cost). Our central structural assumption, that replaces and relaxes specific closure conditions, is the following.

**Definition 1** (Variational Gradient Dominance). We say that $\Pi$ satisfies a $(C_\star, \varepsilon_{\mathrm{vgd}})$-variational gradient dominance (VGD) condition w.r.t. $\mathcal{M}$, if there exist constants $C_\star, \varepsilon_{\mathrm{vgd}} > 0$, such that for any policy $\pi \in \Pi$:

$$V(\pi) - V^\star(\Pi) \leq C_\star \max_{\tilde{\pi} \in \Pi} \langle \nabla V(\pi), \pi - \tilde{\pi} \rangle + \varepsilon_{\mathrm{vgd}}. \quad (2)$$

We note that any policy class satisfies the above conditions with some $C_\star \geq 1, \varepsilon_{\mathrm{vgd}} \leq H$, and that the complete policy class is $(H \|\mu^\star/\rho_0\|_\infty, 0)$-VGD w.r.t. any MDP (see Bhandari & Russo, 2024; Agarwal et al., 2021, and Lemma 16 for completeness). Our main result is the following.

***Theorem*** (informal). Let $\Pi \subset \Delta(\mathcal{A})^{\mathcal{S}}$ be convex and assume it satisfies $(C_\star, \varepsilon_{\mathrm{vgd}})$-VGD w.r.t. $\mathcal{M}$. Suppose further that the actor and critic are approximately optimal up to some error $\varepsilon_{\mathrm{stat}} > 0$. Then, with well tuned $\varepsilon$-greedy exploration and learning rate $\eta$, we have that the PMD method (Equation (1)) converges as follows. With Euclidean regu-

larization, $V(\pi^k) - \min_{\pi^\star \in \Pi} V(\pi^\star) =$

$$O\left( \frac{C_\star^2 H^3 A^{3/2}}{k^{2/3}} + \left( C_\star H + A H^2 k^{1/6} \right) \sqrt{\varepsilon_{\mathrm{stat}}} + \varepsilon_{\mathrm{vgd}} \right),$$

and with negative Entropy regularization, we have that $V(\pi^k) - \min_{\pi^\star \in \Pi} V(\pi^\star) =$

$$O\left( \frac{C_\star^2 H^3 A^{3/2}}{k^{2/7}} + \left( C_\star H + A^2 H^3 k^{4/7} \right) \sqrt{\varepsilon_{\mathrm{stat}}} + \varepsilon_{\mathrm{vgd}} \right),$$

where the big-O only suppresses constant numerical factors.

To obtain our main result, our analysis casts PMD as a proximal point algorithm in a non-Euclidean setting (see Teboulle, 2018 for a review), where the proximal operator uses a regularizer that adapts to *local* smoothness of the objective. As we demonstrate in Lemma 2, the approximation error of the linearization of the objective $V(\cdot)$ at $\pi^k$ can be bounded w.r.t. the local norm $\|\cdot\|_{L^2(\mu^k)}$; crucially, a norm according to which the decision set $\Pi$ has diameter independent of the cardinality of the state-space. This significantly deviates from the commonly used smoothness of the value function w.r.t. the Euclidean norm (Agarwal et al., 2021), which assigns a diameter of $|\mathcal{S}|$ to $\Pi$, and therefore leads to rates that have merit only in tabular environments.

## 1.2. Discussion: VGD vs. Closure

Our work establishes best-in-class convergence subject to the VGD condition presented in the previous section. This is a substantially different starting point than that of the prevalent closure conditions based on the compatible function approximation approach (Sutton et al., 1999) assumed in recent works on parametric policy classes (Agarwal et al., 2021; Yuan et al., 2023; Alfano et al., 2023; Xiong et al., 2024). The assumptions employed in these works fall into two main categories; The first and more general one is that of a bounded *approximation* error (e.g., Alfano et al., 2023; Yuan et al., 2023), which essentially requires that the update step in Equation (1) be close (up to a small error) to the update that would have been performed over the complete policy class $\Pi_{\mathrm{all}} := \Delta(\mathcal{A})^{\mathcal{S}}$. The second is that of bounded *transfer* error (e.g., Agarwal et al., 2021; Yuan et al., 2023), which roughly requires that the update be accurate (up to a small error) when accuracy is measured over the optimal policy occupancy measure. This assumption is commonly employed in the specific log-linear policy class setup; to the best of our knowledge, there do not exist results that employ these conditions in a fully general policy class setting (Agarwal et al., 2021 consider a non-PMD method in a bounded transfer error setup where the policy class satisfies additional smoothness assumptions).

The relation between closure and VGD is subtle, primarily because closure conditions are algorithm dependent. Typically, they relate to one or more of the following three

Table 1: Comparison of assumptions and bounds of representative prior works for PMD with fixed step size. Columns refer to assumptions required either implicitly or explicitly by different works. **VGD** is implied by a natural extension of closure; see Appendix A.1 for further details. The **Realizability** column refers to approximate realizability, which is implied by closure conditions. The **Rate** column suppresses all factors other than $K$, and ignores error floors. • **Closure (perfect):** The policy class is closed to a PMD update up to $\ell_\infty$-norm error. • **Closure (approx):** The policy class is closed to a PMD update up to error that depends on the sampling distribution. • **General dual with EMaP parametrization:** EMaP stands for Exact Mirror and Project; in these works the policy class is induced by a general dual variable parametrization, combined with an operator that performs the mirror and project steps accurately.

| Paper | VGD | $\Pi$ Convexity | Realizability | Closure | Parametric Assumptions | Rate |
|---|---|---|---|---|---|---|
| Xiao (2022); Lan (2023) | Yes | No[a] | Yes | Yes (perfect) | Tabular | $1/K$ |
| Yuan et al. (2023) | Yes[b] | No | Yes | Yes (approx) | Log-linear | $1/K$ |
| Ju & Lan (2022)[c] | Yes | No | Yes | Yes (perfect) | General dual w/ EMaP | $1/\sqrt{K}$ |
| Alfano et al. (2023) | Yes | No | Yes | Yes (approx) | General dual w/ EMaP | $1/K$ |
| **This Work**[d] | Yes | Yes[e] | No | No | No | $1/K^{2/3}$ |

[a] Prior works on the tabular setting typically assume the policy class is complete $\Pi = \Delta(\mathcal{A})^S$, and thus convex. However their arguments extend to the case that $\Pi$ satisfies perfect closure, which eliminates the need for $\Pi$ being convex.

[b] We refer to the bounds obtained by Yuan et al. (2023) subject to bounded approximation error. Yuan et al. (2023) also obtain convergence subject to bounded transfer error — it is unclear to what extent (if at all) bounded transfer error implies VGD.

[c] Ju & Lan (2022) also obtain an $O(1/K)$ rate for regularized PMD.

[d] We report our rate for Euclidean PMD. More generally, our bounds depend on the smoothness of the action regularizer, and dependence on $K$ degrades for non-Euclidean regularizers such as negative entropy.

[e] Assuming only VGD without closure, our analysis requires convexity of $\Pi$. However, in the presence of closure assumptions such as those of Alfano et al. (2023), our analysis does not require convexity of $\Pi$ (see Appendix A.2 for further details).

elements; step-size range, action regularizer, and the particular algorithmic approach employed to solve Equation (1). At the same time, the VGD condition is algorithm independent, as it relates only to the policy class-MDP combination. Nonetheless, as we show in Appendix A.1, a reasonable extension of PMD closure conditions implies variational gradient dominance, effectively establishing PMD closure $\Rightarrow$ VGD. At a high level, this builds on a similar claim from Bhandari & Russo (2024), that closure to policy improvement implies VGD. We further demonstrate in Appendix A.3 that the converse does not hold; that there exist simple examples where the VGD condition holds whereas closure does not take place. We refer to Table 1 for a high level comparison between our work and prior art, and conclude this section with the following additional remarks.

- **Realizability.** Closure conditions generally imply (approximate) realizability, thus under this assumption convergence w.r.t. the true optimal policy $\pi^\star = \arg\min_{\pi \in \Delta(\mathcal{A})^S} V(\pi)$ is possible. We do not assume realizability and therefore prove convergence to the optimal *in-class* policy. Specifically, all prior works prove bounds that only hold in (approximately) realizable settings, while our bounds do not require realizability.

- **Geometric rates.** Table 1 reports rates for fixed step

size PMD. Many prior works that study PMD in the tabular setup or subject to closure conditions establish linear convergence for geometrically increasing step size sequences (Xiao, 2022; Johnson et al., 2023; Yuan et al., 2023; Alfano et al., 2023). We do not expect such rates are possible assuming only VGD. Roughly speaking, the reason these rates are attainable subject to closure is that the algorithm dynamics mimic those of policy iteration in the tabular setting, where convergence is indeed at a linear rate. Assuming only VGD, policy iteration no longer converges, as the policy class loses the favorable structure allowing for convergence of such an aggressive algorithm. This should highlight the value in studying the function approximation setup without closure assumptions.

- **Convexity of the policy class.** Unlike prior works, we consider VGD instead of closure but additionally require convexity of the policy class $\Pi$. However, subject to perfect closure, it can be shown that the iterates of PMD satisfy optimality conditions w.r.t. a convex policy class that contains $\Pi$ (concretely, it will be the complete policy class $\Delta(\mathcal{A})^S$), which is the key element required in our analysis. Thus, our analysis accommodates non convex policy classes as long as perfect closure holds. We refer the reader to Appendix A.2 for a more formal discussion.

## 1.3. Additional Related work

**PMD with non-tabular policy classes.** Most closely related to our work are papers that study convergence of PMD in setups where the policy class is given by function approximators (Vaswani et al., 2022; Ju & Lan, 2022; Grudzien et al., 2022; Alfano et al., 2023; Xiong et al., 2024). The motivation of Alfano et al. (2023); Xiong et al. (2024) is somewhat related to ours but they address a different aspect of the problem in question. These works focus on the approximation errors in the update step (thus essentially assuming closure) and propose algorithmic mechanisms to ensure it is small, but obtain meaningful upper bounds only when it is indeed small w.r.t. the exact steps over the complete policy class (as discussed in the previous section). There is a long line of works on parametric policy classes and specific instantiations of PMD such as the Natural Policy Gradient (NPG; Kakade, 2001); which is the focus of, e.g., Alfano & Rebeschini (2022); Yuan et al. (2023); Cayci et al. (2024) as well as Agarwal et al. (2021). Many works also study convergence dynamics induced by particular policy classes, e.g., Liu et al. (2019); Wang et al. (2020); Liu et al. (2020); we refer the reader to Alfano et al. (2023) for an excellent and more detailed account of these works.

Several prior works have made the observation that PMD is a mirror descent step on the linearization of the value function with a dynamically weighted regularization term (Shani et al., 2020; Tomar et al., 2020; Vaswani et al., 2022; Xiao, 2022), which is the starting point of our work. In particular, this perspective is the focus of Vaswani et al. (2022); however this work did not establish any convergence guarantees.

**PMD in the tabular setting.** The modern analysis approach for PMD in the generic (agnostic to the regularizer) tabular setup is due to Xiao (2022). Additional works that study the tabular setup include Geist et al. (2019); Lan (2023); Johnson et al. (2023); Zhan et al. (2023). As in the function approximation case, many works study convergence of the prototypical PMD instantiation; the NPG or its derivatives TRPO (Schulman et al., 2015) and PPO (Schulman et al., 2017) in tabular or softmax-tabular settings, e.g., Agarwal et al. (2021); Shani et al. (2020); Cen et al. (2022); Bhandari & Russo (2021); Khodadadian et al. (2021; 2022).

**Policy Gradients in parameter space.** There is a rich line of work into policy gradient algorithms that take gradient steps *in parameter space*, both in the tabular and non-tabular setups (Zhang et al., 2020; Mei et al., 2020; 2021; Yuan et al., 2022; Mu & Klabjan, 2024). This class of algorithms are a special case of on-policy PMD only in the case of the direct parametrization, but are not PMD algorithms in general. Most of the results in the case of non-tabular, generic parameterizations characterize convergence in terms

of conditions on the parametric representation. We refer the reader to Yuan et al. (2022) for further review.

**Bregman proximal point methods.** As mentioned, our analysis builds on realizing PMD as an instance of a Bregman proximal point algorithm — roughly, this is a proximal point algorithm (Rockafellar, 1976) in a non-Euclidean setting (see Teboulle (2018) for a review). There are numerous studies that investigate non-Euclidean proximal point methods for both convex and non-convex objective functions (e.g., Tseng, 2010; Ghadimi et al., 2016; Bauschke et al., 2017; Lu et al., 2018; Zhang & He, 2018; Fatkhullin & He, 2024; see also Beck, 2017), although none of them accommodate the particular setup that PMD fits into (see Appendix E for details). Our analysis for the proximal point method presented in Section 3.2 is mostly inspired by the work of Xiao (2022); specifically, their upper bounds for projected gradient descent, where they apply a proximal point analysis in the euclidean setting.

## 2. Preliminaries

**Discounted MDPs.** A discounted MDP $\mathcal{M}$ is defined by a tuple $\mathcal{M} = (\mathcal{S}, \mathcal{A}, \mathbb{P}, r, \gamma, \rho_0)$, where $\mathcal{S}$ denotes the state-space, $\mathcal{A}$ the action set, $\mathbb{P}\colon \mathcal{S} \times \mathcal{A} \to \Delta(\mathcal{S})$ the transition dynamics, $r\colon \mathcal{S} \times \mathcal{A} \to [0, 1]$ the reward function, $0 < \gamma < 1$ the discount factor, $H \coloneqq \frac{1}{1-\gamma}$ the effective horizon, and $\rho_0 \in \Delta(\mathcal{S})$ the initial state distribution. For notational convenience, for $s, a \in \mathcal{S} \times \mathcal{A}$ we let $\mathbb{P}_{s,a} \coloneqq \mathbb{P}(\cdot \mid s, a) \in \Delta(\mathcal{S})$ denote the next state probability measure.

We assume the action set is finite with $A \coloneqq |\mathcal{A}|$, and identify $\mathbb{R}^A$ with $\mathbb{R}^{\mathcal{A}}$. We additionally assume, for clarity of exposition and in favor of simplified technical arguments, that the state space is finite with $S \coloneqq |\mathcal{S}|$, and identify $\mathbb{R}^S$ with $\mathbb{R}^{\mathcal{S}}$. We emphasize that all our arguments may be extended to the infinite state-space setting with additional technical work. An agent interacting with the MDP is modeled by a policy $\pi\colon \mathcal{S} \to \Delta(\mathcal{A})$, for which we let $\pi_s \in \Delta(\mathcal{A}) \subset \mathbb{R}^A$ denote the action probability vector at $s$ and $\pi_{s,a} \in [0, 1]$ denote the probability of taking action $a$ at $s$. We denote the *value* of $\pi$ when starting from a state $s \in \mathcal{S}$ by $V_s(\pi)$:

$$V_s(\pi) \coloneqq \mathbb{E}\left[\sum_{t=0}^{\infty} \gamma^t r(s_t, a_t) \mid s_0 = s, \pi\right],$$

and more generally for any $\rho \in \Delta(\mathcal{S})$, $V_\rho(\pi) \coloneqq \mathbb{E}_{s \sim \rho} V_s(\pi)$. When the subscript is omitted, $V(\pi)$ denotes value of $\pi$ when starting from the initial state distribution $\rho_0$:

$$V(\pi) \coloneqq V_{\rho_0}(\pi) = \mathbb{E}\left[\sum_{t=0}^{\infty} \gamma^t r(s_t, a_t) \mid s_0 \sim \rho_0, \pi\right].$$

For any state action pair $s, a \in \mathcal{S} \times \mathcal{A}$, the action-value function of $\pi$, or $Q$-function, measures the value of $\pi$ when

starting from $s$, taking action $a$, and then following $\pi$ for the reset of the interaction:

$$Q_{s,a}^{\pi} \coloneqq \mathbb{E}\left[\sum_{t=0}^{\infty} \gamma^t r(s_t, a_t) \mid s_0 = s, a_0 = a, \pi\right]$$

We further denote the discounted state-occupancy measure of $\pi$ induced by any start state distribution $\rho \in \Delta(\mathcal{S})$ by $\mu_\rho^\pi$:

$$\mu_\rho^\pi(s) \coloneqq (1 - \gamma) \sum_{t=0}^{\infty} \gamma^t \Pr(s_t = s \mid s_0 \sim \rho, \pi).$$

It is easily verified that $\mu^\pi \in \Delta(\mathcal{S})$ is indeed a state probability measure. In the sake of brevity, we take the MDP true start state distribution $\rho_0$ as the default in case one is not specified:

$$\mu^\pi \coloneqq \mu_{\rho_0}^\pi. \tag{3}$$

**Learning objective.** In the conventional formulation of MDPs, the objective is to maximize the discounted total reward, i.e., $\max_\pi V(\pi)$. In this paper, we follow Xiao (2022) and adopt a minimization formulation in order to better align with conventions in the optimization literature. To this end, we regard each $r(s, a) \in [0, 1]$ as a value measuring regret, or cost, rather than reward. Given any reward function $r$, we may reset $r(s, a) \leftarrow 1 - r(s, a)$ for all $s, a \in \mathcal{S} \times \mathcal{A}$ to transform it into a regret function. With this in mind, we consider the problem of finding an approximately optimal policy within a given policy class $\Pi \subset \Delta(\mathcal{A})^{\mathcal{S}}$:

$$\arg\min_{\pi \in \Pi} V(\pi). \tag{4}$$

To avoid ambiguity, we denote the optimal value attainable by an in-class policy (a solution to Equation (4)) by $V^\star(\Pi)$, and the optimal value attainable by any policy by $V^\star$:

$$V^\star(\Pi) \coloneqq \arg\min_{\pi^\star \in \Pi} V(\pi^\star); \quad V^\star \coloneqq \arg\min_{\pi^\star \in \Delta(\mathcal{A})^{\mathcal{S}}} V(\pi^\star). \tag{5}$$

We note that we do not make any explicit structural assumptions about $\mathcal{M}$. We will however make some assumptions about the policy class $\Pi$, which will be made clear in the statements of our theorems.

### 2.1. Problem Setup

In this work, we focus on the PMD method Algorithm 1 for solving Equation (4) in the case that the policy class is non-complete, $\Pi \neq \Delta(\mathcal{A})^{\mathcal{S}}$. In each iteration, PMD solves a stochastic optimization sub-problem formed by an estimate of the current policy $Q$-function and a Bregman divergence term which is defined below.

---

**Algorithm 1** Policy Mirror Descent (on-policy)

**Input:** learning rate $\eta > 0$, regularizer $R \colon \mathbb{R}^{\mathcal{A}} \to \mathbb{R}$
Initialize $\pi^1 \in \Pi$
**for** $k = 1$ to $K$ **do**
  Set $\mu^k \coloneqq \mu^{\pi^k}$; $\widehat{Q}^k \coloneqq \widehat{Q}^{\pi^k}$.
  $\pi^{k+1} \leftarrow \arg\min_{\pi \in \Pi} \mathbb{E}_{s \sim \mu^k}\left[H\left\langle \widehat{Q}_s^k, \pi_s \right\rangle + \frac{1}{\eta} B_R(\pi_s, \pi_s^k)\right]$
**end for**

---

**Definition 2** (Bregman divergence). Given a convex differentiable regularizer $R \colon \mathbb{R}^{\mathcal{A}} \to \mathbb{R}$, the Bregman divergence w.r.t. $R$ is:

$$B_R(u, v) \coloneqq R(u) - R(v) - \langle \nabla R(v), u - v \rangle.$$

Throughout, we make the following assumptions regarding the solutions to the sub-problems and the $Q$-function estimates Algorithm 1.

**Assumption 1** (Sub-problem optimization oracle). We assume that for all $k$, $\pi^{k+1}$ is approximately optimal, in the sense that constrained optimality conditions hold up to error $\varepsilon_{\text{act}}$:

$$\forall \pi \in \Pi, \left\langle \nabla \phi_k(\pi^{k+1}), \pi - \pi^{k+1} \right\rangle \geq -\varepsilon_{\text{act}},$$

where $\phi_k(\pi) \coloneqq \mathbb{E}_{s \sim \mu^k}\left[H\left\langle \widehat{Q}_s^k, \pi_s \right\rangle + \frac{1}{\eta} B_R(\pi_s, \pi_s^k)\right]$.

**Assumption 2** (Q-function oracle). We assume that for all $\pi$,

$$\mathbb{E}_{s \sim \mu^\pi}\left[\left\|\widehat{Q}_s^\pi - Q_s^\pi\right\|_2^2\right] \leq \varepsilon_{\text{crit}}.$$

We remark that our results can be easily adapted to somewhat weaker conditions on the critic error; we defer the discussion to Appendix B.1.

**Additional notation.** Given a state probability measure $\mu \in \Delta(\mathcal{S})$ and an action space norm $\|\cdot\|_\circ \colon \mathbb{R}^A \to \mathbb{R}$, we define the induced state-action weighted $L^p$ norm $\|\cdot\|_{L^p(\mu),\circ} \colon \mathbb{R}^{SA} \to \mathbb{R}$ as follows:

$$\|u\|_{L^p(\mu),\circ} \coloneqq \left(\mathbb{E}_{s \sim \mu} \|u_s\|_\circ^p\right)^{1/p}.$$

For any norm $\|\cdot\|$, we let $\|\cdot\|^*$ denote its dual. When discussing a generic norm and there is no risk of confusion, we may use $\|\cdot\|_*$ to refer to its dual. We repeat the following notation that is used throughout the paper for convenience:

$$\mu^\pi \coloneqq \mu_{\rho_0}^\pi, \quad S \coloneqq |\mathcal{S}|, \quad A \coloneqq |\mathcal{A}|, \quad H \coloneqq \frac{1}{1 - \gamma}.$$

### 2.2. Optimization preliminaries

We proceed with several basic definitions before concluding the setup.

**Definition 3** (Lipschitz Gradient). We say a function $h: \Omega \to \mathbb{R}$, $\Omega \subseteq \mathbb{R}^d$ has an $L$-Lipschitz gradient or is $L$-smooth w.r.t. a norm $\|\cdot\|$ if for all $x, y \in \Omega$:

$$\|\nabla h(x) - \nabla h(y)\|_* \leq L \|x - y\|.$$

**Definition 4** (Gradient Dominance). We say $f: \mathcal{X} \to \mathbb{R}$ satisfies the variational gradient dominance condition with parameters $(C_\star, \delta)$, or that $f$ is $(C_\star, \delta)$-VGD, if here exist constants $C_\star, \delta > 0$, such that for any $x \in \mathcal{X}$, it holds that:

$$f(x) - \arg\min_{x^\star \in \mathcal{X}} f(x^\star) \leq C_\star \max_{\tilde{x} \in \mathcal{X}} \langle \nabla f(x), x - \tilde{x} \rangle + \delta.$$

**Definition 5** (Local Norm). We define a *local* norm over a set $\mathcal{X} \subseteq \mathbb{R}^d$ by a mapping $x \mapsto \|\cdot\|_x$ such that $\|\cdot\|_x$ is a norm for all $x \in \mathcal{X}$. We may denote a local norm by $\|\cdot\|_{(\cdot)}$ or by $x \mapsto \|\cdot\|_x$.

**Definition 6** (Local Smoothness). We say $f: \mathcal{X} \to \mathbb{R}$ is $\beta$-*locally* smooth w.r.t. a local norm $x \mapsto \|\cdot\|_x$ if for all $x, y \in \mathcal{X}$:

$$|f(y) - f(x) - \langle \nabla f(x), y - x \rangle| \leq \frac{\beta}{2} \|y - x\|_x^2.$$

## 3. Best-in-class Convergence of Policy Mirror Descent

In this section, we present our main results which establish convergence rates for the PMD method in the non-complete class setting we consider. Our main theorem, given below, provides convergence rates for two classic instantiations of PMD; with Euclidean regularization and negative entropy regularization. Our results require that $\epsilon$-greedy exploration be incorporated into the policy class. To that end, let $\Pi^\epsilon$ denote the policy class obtained by adding $\epsilon$-greedy exploration to $\Pi$:

$$\Pi^\epsilon := \{(1 - \epsilon)\pi + \epsilon u \mid \pi \in \Pi\}, \text{ where } u_{s,a} \equiv 1/A.$$

We have the following.

**Theorem 1.** *Let $\Pi \subset \Delta(\mathcal{A})^S$ be convex and assume it is $(C_\star, \varepsilon_{\text{vgd}})$-VGD w.r.t. $\mathcal{M}$. Consider the on-policy PMD method Algorithm 1 when run over $\Pi^{\varepsilon_{\text{expl}}}$. Then, assuming $\varepsilon_{\text{act}} + \varepsilon_{\text{crit}} \leq \varepsilon_{\text{stat}}$, and with proper tuning of $\eta, \varepsilon_{\text{expl}}$, it holds that:*

*i. If $R(p) = \frac{1}{2} \|p\|_2^2$ is the Euclidean action-regularizer, we have $V(\pi^K) - V^\star(\Pi) =$*

$$O\left(\frac{C_\star^2 A^{3/2} H^3}{K^{2/3}} + \left(C_\star H + A H^2 K^{1/6}\right)\sqrt{\varepsilon_{\text{stat}}} + \varepsilon_{\text{vgd}}\right)$$

*ii. If $R(p) = \sum_i p_i \log p_i$ is the negative entropy action-regularizer, we have $V(\pi^K) - V^\star(\Pi) =$*

$$O\left(\frac{C_\star^2 A^{3/2} H^3}{K^{2/7}} + \left(C_\star H + A^2 H^3 K^{4/7}\right)\sqrt{\varepsilon_{\text{stat}}} + \varepsilon_{\text{vgd}}\right).$$

*In both cases, big-O notation suppresses only constant factors.*

To our knowledge, Theorem 1 is the first result to establish best-in-class convergence (at any rate) of PMD without closure conditions. Two additional comments are in order: (1) Our current analysis technique requires the action regularizer to be smooth. This is also the source of the degraded rate in the negative entropy case. (2) The greedy exploration stems from the smoothness parameter we establish for the value function, and leads to worse rates in the Euclidean case (for negative entropy, it actually implies smoothness of the regularizer, though this is not the primary reason for which it is introduced). We discuss this point further in Section 3.1.

**Analysis overview.** The analysis leading up to Theorem 1 builds on casting PMD as an instance of a Bregman proximal point (or equivalently, a mirror descent) algorithm. This follows by demonstrating PMD proceeds by optimizing subproblems formed by linear approximations of the value function and a proximity term that adapts to *local* smoothness of the objective, as measured by the norm induced by the current policy occupancy measure.

In fact, it has already been previously observed (e.g., Shani et al., 2020; Xiao, 2022) that the on-policy PMD update step is completely equivalent to a mirror descent step w.r.t. the value function gradient equipped with a dynamically weighted proximity term. For any two policies $\pi$ and $\pi^k$, by the policy gradient theorem (Sutton et al., 1999, see also Lemma 15 in Appendix D.1):

$$\mathbb{E}_{s \sim \mu^k}\left[H\langle Q_s^k, \pi_s \rangle + \frac{1}{\eta}B_R(\pi_s, \pi_s^k)\right]$$

$$= \langle \nabla V(\pi^k), \pi \rangle + \frac{1}{\eta}B_{\pi^k}(\pi, \pi^k), \qquad (6)$$

where we denote $\mu^k := \mu^{\pi^k}$, $Q^k := Q^{\pi^k}$, and $B_{\pi^k}(u, v) := \mathbb{E}_{s \sim \mu^k} B_R(u_s, v_s)$. However, these prior observations did not yield new convergence results, as the algorithm in question significantly deviates from a standard instantiation of mirror descent; a priori, it is unclear how the regularizer associated with $B_{\pi^k}$ relates to the objective in optimization terms.

The high level components of our analysis are outlined next. In Section 3.1 we establish local smoothness of the value function (Lemma 2), which is the key element in establishing convergence of PMD through a proximal point algorithm perspective. Then, in Section 3.2 we introduce the optimization setup that accommodates proximal point methods that adapt to local smoothness of the objective, and present the convergence guarantees for this class of algorithms. Finally, we return to prove Theorem 1 in Appendix D.3, where we apply both Lemma 2 and the result of Section 3.2 to establish convergence of PMD.

## 3.1. Local smoothness of the value function

The principal element of our approach builds on smoothness of the value function w.r.t. the local norm induced by the occupancy measure of the policy at which we take the linear approximation, given by the below lemma. We defer the proof to Appendix D.2.

**Lemma 2.** *Let* $\pi\colon \mathcal{S} \to \Delta(\mathcal{A})$ *be any policy such that* $\epsilon := \min_{s,a} \{\pi_{sa}\} > 0$. *Then, for any* $\tilde{\pi} \in \mathcal{S} \to \Delta(\mathcal{A})$, *we have:*

$$
|V(\tilde{\pi}) - V(\pi) - \langle \nabla V(\pi), \tilde{\pi} - \pi \rangle|
$$
$$
\leq \min\left\{ \frac{H^3}{\sqrt{\epsilon}} \|\tilde{\pi} - \pi\|^2_{L^2(\mu^\pi),1}, \frac{AH^3}{\sqrt{\epsilon}} \|\tilde{\pi} - \pi\|^2_{L^2(\mu^\pi),2} \right\}.
$$

It is instructive to consider Lemma 2 in the context of the more standard non-weighted $L^2$ smoothness property established in Agarwal et al. (2021).

- **Dependence on $S$:** The standard $L^2$ smoothness leads to rates that scale with $\|\pi^1 - \pi^\star\|_2$, which scales with $S$ in general. Indeed, prior works that exploit smoothness of the value function (e.g., Agarwal et al., 2021; Xiao, 2022) derive bounds for PGD (i.e., mirror descent with non-local, euclidean regularization) that do in fact hold in the setting we consider here, but inevitably lead to convergence rates that scale with the cardinality of the state-space. This is while the diameter assigned to the decision set $\Pi$ by $\|\cdot\|_{L^2(\mu^\pi),\circ}$, for any $\pi$, depends only on the diameter assigned to $\Delta(\mathcal{A})$ by $\|\cdot\|_\circ$, and thus is independent of $S$.

- **Relation to PMD:** The standard $L^2$ smoothness does not naturally integrate with the PMD framework, and leads to algorithms (such as vanilla projected gradient descent) where the update step cannot be framed as a solution to a stochastic optimization problem induced by some policy occupancy measure. As such, these do not admit a formulation that is easily implemented in practical applications.

- **Smoothness parameter:** The smoothness parameter in Lemma 2 depends on the minimum action probability assigned by the policy at which we linearize the value function (and as we discuss in Appendix B.2, this is not an artifact of our analysis). A simple resolution for this is given by adding $\epsilon$-greedy exploration. Notably, the relatively large $O(1/\sqrt{\epsilon})$ smoothness constant ultimately leads to a rate that is worse than the $O(1/K)$ achievable with the standard $L^2$ smoothness (but that crucially, does not scale with $S$).

## 3.2. Digression: Constrained non-convex optimization for locally smooth objectives

In this section, we consider the constrained optimization problem:

$$
\min_{x \in \mathcal{X}} f(x), \tag{7}
$$

where the decision set $\mathcal{X} \subseteq \mathbb{R}^d$ is convex and endowed with a local norm $x \mapsto \|\cdot\|_x$ (see Definition 5), and $f$ is differentiable over an open domain that contains $\mathcal{X}$. We assume access to the objective is granted through an approximate first order oracle, as defined next.

**Assumption 3.** We have first order access to $f$ through an $\varepsilon_\nabla$-approximate gradient oracle; For all $x \in \mathcal{X}$, we have

$$
\left\| \widehat{\nabla} f(x) - \nabla f(x) \right\|^*_x \leq \varepsilon_\nabla \leq 1.
$$

Theorem 3 given below establishes convergence rates for the algorithm we describe next. Given an initialization $x_1 \in \mathcal{X}$, learning rate $\eta > 0$, and local regularizer $R_x\colon \mathbb{R}^d \to \mathbb{R}$ for all $x \in \mathcal{X}$, iterate for $k = 1, \ldots, K$:

$$
x_{k+1} = \arg\min_{y \in \mathcal{X}} \left\{ \left\langle \widehat{\nabla} f(x), y \right\rangle + \frac{1}{\eta} B_{R_x}(y, x) \right\}. \tag{8}
$$

The above algorithm can be viewed as either a mirror descent algorithm (Nemirovskij & Yudin, 1983; Beck & Teboulle, 2003) or a proximal point algorithm (Rockafellar, 1976) in a non-Euclidean setup (see Teboulle, 2018 for a review), where the non-smooth term is the decision set indicator function. Our analysis (detailed in Appendix E) hinges on a descent property of the algorithm, thus naturally takes the proximal point perspective. We prove the following.

**Theorem 3.** *Suppose that* $f$ *is* $(C_\star, \varepsilon_{\mathrm{vgd}})$*-VGD as per Definition 4, and that* $f^\star := \min_{x \in \mathcal{X}} f(x) > -\infty$. *Assume further that:*

*(i) The local regularizer* $R_x$ *is* 1*-strongly convex and has an* $L$*-Lipschitz gradient w.r.t.* $\|\cdot\|_x$ *for all* $x \in \mathcal{X}$.

*(ii) For all* $x \in \mathcal{X}$, $\max_{u,v \in \mathcal{X}} \|u - v\|_x \leq D$, *and* $\|\nabla f(x)\|^*_x \leq M$.

*(iii)* $f$ *is* $\beta$*-locally smooth w.r.t.* $x \mapsto \|\cdot\|_x$.

*Then, assuming* $x^{k+1}$ *are* $\varepsilon_{\mathrm{opt}}$*-approximately optimal (in the same sense of Assumption 1), the proximal point algorithm Equation (8) has the following guarantee when* $\eta \leq 1/(2\beta)$:

$$
f(x_{K+1}) - f^\star = O\left( \frac{C_\star^2 L^2 c_1^2}{\eta K} + \mathcal{E}_{\mathrm{err}} + \varepsilon_{\mathrm{vgd}} \right)
$$

*where* $c_1 := D + \eta M$ *and*

$$
\mathcal{E}_{\mathrm{err}} := \left( C_\star D + c_1 L^2 \right) \varepsilon_\nabla + C_\star \varepsilon_{\mathrm{opt}} + c_1 L \sqrt{\varepsilon_{\mathrm{opt}}/\eta}.
$$

*where* $c_1 := D + \eta M$.

The proof of Theorem 3 as well as additional technical details for this section are provided in Appendix E.

### 3.3. Proof of main result

To prove our main result, we begin with a lemma that essentially "maps" the PMD setup into the optimization framework of Section 3.2. The proof consists of showing that the appropriate assumptions on actor, critic, and action regularizer translate to the conditions of Theorem 3 for locally smooth optimization.

**Lemma 4.** *Let $\Pi$ be a convex policy class that is $(C_\star, \varepsilon_{\mathrm{vgd}})$-VGD w.r.t. the MDP $\mathcal{M}$. Consider the on-policy PMD method Algorithm 1, and assume that the following conditions hold:*

*(i) $R\colon \mathbb{R}^{\mathcal{A}} \to \mathbb{R}$ is 1-strongly convex and has an L-Lipschitz gradient w.r.t. an action-space norm $\|\cdot\|_\circ$.*

*(ii) $\max_{p,q \in \Delta(\mathcal{A})} \|p - q\|_\circ \leq D$, and $\|Q_s^\pi\|_\circ^* \leq M$ for all $s \in \mathcal{S}, \pi \in \Pi$.*

*(iii) The value function is $\beta$-locally smooth over $\Pi$ w.r.t. the local norm $\|\cdot\|_\pi := \|\cdot\|_{L^2(\mu^\pi),\circ}$.*

*Then, we have the following guarantee:*

$$V(\pi^K) - V^\star(\Pi) = O\left(\frac{C_\star^2 L^2 c_1^2}{\eta K} + \mathcal{E}_{\mathrm{stat}} + \varepsilon_{\mathrm{vgd}}\right)$$

*where $c_1 := D + \eta H M$, and*

$$\mathcal{E}_{\mathrm{stat}} = \left(C_\star D + c_1 L^2\right) H \sqrt{\varepsilon_{\mathrm{crit}}} + C_\star \varepsilon_{\mathrm{act}} + c_1 L \sqrt{\varepsilon_{\mathrm{act}}/\eta}.$$

For $\mu \in \mathbb{R}^S, Q \in \mathbb{R}^{SA}$, we define the state to state-action element-wise product $\mu \circ Q \in \mathbb{R}^{SA}$ by $(\mu \circ Q)_{s,a} := \mu(s)Q_{s,a}$. Observe that for all $k$, it holds that

$$\mathbb{E}_{s\sim\mu^k}\left[H\left\langle \widehat{Q}_s^k, \pi_s\right\rangle + \frac{1}{\eta}B_R(\pi_s, \pi_s^k)\right]$$
$$= \left\langle \widehat{\nabla}V(\pi^k), \pi\right\rangle + \frac{1}{\eta}B_{\pi^k}(\pi, \pi^k),$$

with: $B_{\pi^k}(\pi, \tilde{\pi}) := \mathbb{E}_{s\sim\mu^k}B_R(\pi_s, \tilde{\pi}_s)$, $\widehat{\nabla}V(\pi) := H\mu^\pi \circ \widehat{Q}^\pi$. Next, we demonstrate PMD is an instance of the optimization algorithm Equation (8), and verify that all of the conditions in Theorem 3 hold w.r.t. the local norm $\pi \mapsto \|\cdot\|_{L^2(\mu^\pi),\circ}$. First, to see that the gradient error is bounded by $H\sqrt{\varepsilon_{\mathrm{crit}}}$, observe:

$$\left\|\widehat{\nabla}V(\pi) - \nabla V(\pi)\right\|_{L^2(\mu^\pi),\circ}^* = \left\|\mu^\pi \circ \left(\widehat{Q}^\pi - Q^\pi\right)\right\|_{L^2(\mu^\pi),\circ}^*$$
$$= \sqrt{\mathbb{E}_{s\sim\mu^\pi}\left(\left\|\widehat{Q}^\pi - Q^\pi\right\|_\circ^*\right)^2}$$
$$\leq \sqrt{\varepsilon_{\mathrm{crit}}},$$

where second inequality follows from Lemma 12 and the inequality from Assumption 2. Further:

1. By a simple relation (Lemma 13) between $R$ and the state-action it regularizer it induces defined below,

$$R_{\pi^k}(\pi) := \mathbb{E}_{s\sim\mu^k}R(\pi_s),$$

we have that $B_{\pi^k}(\cdot, \cdot)$ is the Bregman divergence of $R_{\pi^k}$, and further using (i) that $R_{\pi^k}$ is 1-strongly convex and has an $L$-Lipschitz gradient w.r.t. $\|\cdot\|_{L^2(\mu^k),\circ}$.

2. For all $\pi, \pi', \tilde{\pi}$, by (ii),

$$\|\pi' - \tilde{\pi}\|_{L^2(\mu^\pi),\circ} = \sqrt{\mathbb{E}_{s\sim\mu^\pi}\|\pi'_s - \tilde{\pi}_s\|^2} \leq D.$$

In addition by (ii) and the dual norm expression (Lemma 12), for any $\pi$:

$$\|\nabla V(\pi)\|_{L^2(\mu^\pi),\circ}^* = H\|\mu^\pi \circ Q^\pi\|_{L^2(\mu^\pi),\circ}^*$$
$$= H\sqrt{\mathbb{E}_{s\sim\mu^\pi}\left(\|Q_s^\pi\|_\circ^*\right)^2} \leq HM.$$

3. Finally, the objective is $\beta$-locally smooth by assumption (iii).

The result now follows from Theorem 3. $\qquad\square$

We conclude with a proof sketch of Theorem 1 for the Euclidean case; the full technical details are provided in Appendix D.3.

*Proof sketch of Theorem 1 (Euclidean case).* The first step is showing that the $\varepsilon_{\mathrm{expl}}$-greedy exploration introduces an error term that scales with $\delta := \varepsilon_{\mathrm{expl}}C_\star H^2 A$ (see Lemma 19). This implies that $\Pi^{\varepsilon_{\mathrm{expl}}}$ is $(C_\star, \varepsilon_{\mathrm{vgd}} + \delta)$-VGD w.r.t. $\mathcal{M}$. In addition, by definition of $\Pi^{\varepsilon_{\mathrm{expl}}}$ we have $\min_{s,a}\{\pi_{s,a}\} \geq \varepsilon_{\mathrm{expl}}/A$ for all $\pi \in \Pi^{\varepsilon_{\mathrm{expl}}}$. We now argue the following:

1. The action regularizer $R(p) = \frac{1}{2}\|p\|_2^2$ is 1-strongly convex and has 1-Lipschitz gradient w.r.t. $\|\cdot\|_2$.

2. $\forall s, \|\pi_s - \tilde{\pi}_s\|_2 \leq D = 2, \|Q_s\|_2 \leq M = \sqrt{AH}$.

3. By Lemma 2, the value function is $\left(\beta := \frac{A^{3/2}H^3}{\sqrt{\varepsilon_{\mathrm{expl}}}}\right)$-locally smooth w.r.t. $\pi \mapsto \|\cdot\|_{L^2(\mu^\pi),2}$.

The result now follows from Lemma 4 with $\eta = 1/(2\beta)$ and $\varepsilon_{\mathrm{expl}} = K^{-2/3}$. $\qquad\square$

# 4. Conclusions and Outlook

In this work, we introduced a novel theoretical framework and established best-in-class convergence of PMD for general policy classes, subject to an algorithm independent variational gradient dominance condition instead of a closure condition. In addition, we discussed the relation between VGD and closure thoroughly, and demonstrated closure implies VGD but not the other way around (Section 1.2 and Appendix A). We conclude by outlining two directions for valuable (in our view) future research.

- **$\epsilon$-greedy exploration.** Our approach builds on ensuring descent on each iteration, which we establish by demonstrating local smoothness holds *globally*, for any reference policy $\tilde{\pi}$. As we discuss in Appendix B.2, it seems that this technique cannot yield better results. However, when the multiplicative ratio $|\pi_{s,a}/\tilde{\pi}_{s,a} - 1|$ is bounded, arguments similar to those given in Appendix D.2 demonstrate a somewhat weaker notion of smoothness — but *without* dependence on the exploration parameter. Furthermore, an analysis approach that combines with the classic mirror descent analysis might do without the per iteration descent property.

- **Non-smooth action regularizers.** Our approach encounters an obstacle that seems related to existing techniques for non-convex, non-Euclidean proximal point methods, which leads to the requirement of a smooth regularizer. This is also the source of the degraded rate in the negative entropy case. Progress can be made by either advancing state-of-the-art in this area of optimization (or showing the limitation is inherent to the setup), or alternatively exploiting additional structure specific to the value function.

# Acknowledgements

This project has received funding from the European Research Council (ERC) under the European Union's Horizon 2020 research and innovation program (grant agreements No. 101078075; 882396). Views and opinions expressed are however those of the author(s) only and do not necessarily reflect those of the European Union or the European Research Council. Neither the European Union nor the granting authority can be held responsible for them. This work received additional support from the Israel Science Foundation (ISF, grant numbers 3174/23; 1357/24), and a grant from the Tel Aviv University Center for AI and Data Science (TAD). This work was partially supported by the Deutsch Foundation.

# Impact Statement

This paper presents work whose goal is to advance the field of Machine Learning. There are many potential societal consequences of our work, none which we feel must be specifically highlighted here.

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

# A. Variational Gradient Dominance and Closure Conditions

In this section, we include detailed discussions regarding the VGD and closure conditions. In Appendix A.1, we demonstrate closure $\implies$ VGD; In Appendix A.3, we show that VGD $\not\Rightarrow$ closure — we present a simple example where VGD holds, but closure doesn't and furthermore that bounds of prior works fail to capture convergence of PMD; Finally, in Appendix A.4, we conclude with several general remarks.

## A.1. Closure implies VGD

In this section, we provide formal proofs that closure conditions employed by prior works imply the VGD condition. Throughout this section, in favor of a simpler comparison, we assume the critic and actor errors are zero, i.e., all algorithms have access to exact action-value functions, and $\varepsilon_{\text{act}} = 0$. We introduce the following, slightly extended version of the VGD condition.

**Definition 7.** We say a policy class $\Pi$ satisfies $(C_\star, \varepsilon_{\text{vgd}}; v^\star)$-VGD if for all $\pi \in \Pi$:

$$V(\pi) - v^\star \le C_\star \max_{\tilde{\pi} \in \Pi} \langle \nabla V(\pi), \pi - \tilde{\pi} \rangle + \varepsilon_{\text{vgd}}.$$

The above extension of the VGD assumption enables a clearer comparison with prior works. As closure implies (approximate) realizability, prior works obtain bounds w.r.t. the optimal (potentially out-of-class) value function $V^\star = \min_{\pi \in \Delta(\mathcal{A})^S} V(\pi)$. Our original VGD condition Definition 1 is stated with the reasonable $v^\star = V^\star(\Pi)$ choice, however our bounds hold just the same under the assumption VGD holds for other $v^\star$ (such as $v^\star = V^\star$).

As we show next, both Yuan et al. (2023)[1](see Lemma 6) and Alfano et al. (2023) (see Lemma 8) adopt assumptions that imply their policy classes satisfy Definition 7 with suitable parameters $C_\star, \varepsilon_{\text{vgd}}$ and $v^\star = V^\star$. Notably, the error floors in their convergence results are indeed precisely (up to constant factors) $\varepsilon_{\text{vgd}}$. We note the implication we establish is not "perfect", to make the argument we need slight variations of the original algorithm dependent conditions — our goal here is to highlight the strong relation between the two assumptions. Before proceeding, we specifically note the following:

- To simplify presentation, we consider closure assumptions (bounded approximation error, concentrability, and distribution mismatch) globally, rather than on the specific iterates selected by the algorithm. However, the same arguments can be made iterate specific, which would lead to VGD conditions on the specific iterates, which is indeed all that is required by our analyses.

- The concentrability assumptions employed by Yuan et al. (2023); Alfano et al. (2023) relate to the current policy $\pi^k$ and the next one $\pi^{k+1}$. The direct global extension of this condition would concern a policy $\pi$ and a policy $\pi^+$ selected by a step of the algorithm with the given step size and regularizer. Our proof requires $\pi^+$ to be selected differently (e.g., with a different step size choice), which leads to a concentrability assumption that relates to a different $\pi^+$ than the original ones. In this sense, the assumption we make here is a different one, but still qualitatively similar. Again, to simplify presentation we assume stronger concentrability in the lemma statements where $\pi^+$ may be arbitrary, but this can be relaxed as explained above. In addition, our concentrability requires the sampling distribution $v^k$ to support *the current* occupancy measure $\mu^k$ rather than the next one $\mu^{k+1}$. This may actually be considered a weaker assumption than the original one, as the next policy is only determined after performing the step that uses $v^k$. Further, we may always simply select $v^k = \mu^k \circ \pi^k$ to obtain optimal support for $\mu^k$.

- In Lemma 8, we prove that when (the natural extension of) closure assumptions of Alfano et al. (2023) hold for Euclidean regularization, VGD holds as well. The claim can be extended to other regularizers with the price of additional regularity assumptions. Regardless, the bounded approximation error assumption of Alfano et al. (2023) may alternatively be interpreted as a bound on the statistical error, in which case the policy class operated over is $(v_\star, 0; V^\star)$-VGD; we provide further details in Appendix A.2.

- The work of Bhandari & Russo (2024) demonstrated that closure to policy improvement implies VGD, and further observed there is also a connection between bounded approximation error and VGD (Lemma 16, Appendix B in their work). The arguments we give below may be considered a generalization of those in Bhandari & Russo (2024), strengthening the connection between closure and VGD.

---

[1]We refer to their results based on bounded approximation error.

**Lemma 5** (Generic closure $\implies$ VGD). *Let $\Pi$ be a policy class, $\pi \in \Pi$ a policy, and $v \in \Delta(\mathcal{S} \times \mathcal{A})$ a state-action probability measure. Suppose there exists $\pi^+ \in \Pi$ such that:*

$$\mathbb{E}_{s \sim \mu^\pi} \left\langle \widehat{Q}_s^\pi, \pi_s^+ \right\rangle \leq \mathbb{E}_{s \sim \mu^\pi} \min_a \widehat{Q}_{s,a}^\pi + \varepsilon_{\text{greedy}},$$

$$where \; \mathbb{E}_{s,a \sim v} \left[ \left( \widehat{Q}_{s,a}^\pi - Q_{s,a}^\pi \right)^2 \right] \leq \varepsilon_{\text{approx}},$$

*and further:*

$$\mathbb{E}_{s,a \sim v} \left[ \left( \frac{\tilde{\mu}(s) \tilde{\pi}_{s,a}}{v(s,a)} \right)^2 \right] \leq C_v, \qquad \text{(}v\text{-concentrability)}$$

*For $\tilde{\pi} \in \{\pi, \pi^+, \pi^\star\}, \tilde{\mu} \in \{\mu^\pi, \mu^\star\}$, where $\pi^\star = \arg\min_{\pi \in \Delta(\mathcal{A})^\mathcal{S}} V(\pi), \mu^\star = \mu^{\pi^\star}$. Then, it holds that*

$$V(\pi) - V^\star \leq \left\| \frac{\mu^\star}{\mu^\pi} \right\|_\infty \max_{\tilde{\pi} \in \Pi} \langle \nabla V(\pi), \pi - \tilde{\pi} \rangle + H \left\| \frac{\mu^\star}{\mu^\pi} \right\|_\infty \left( \varepsilon_{\text{greedy}} + 4\sqrt{C_v \varepsilon_{\text{approx}}} \right).$$

*Proof.* We first establish bounds on approximation error terms, then proceed to leverage the approximate greedification assumption to establish VGD.

**Approximation error.** For any policy $\tilde{\pi}$ and state-occupancy $\tilde{\mu}$, , we have:

$$\mathbb{E}_{s \sim \tilde{\mu}} \left\langle Q_s^\pi - \widehat{Q}_s^\pi, \tilde{\pi}_s \right\rangle = \left\langle Q^\pi - \widehat{Q}^\pi, \tilde{\mu} \circ \tilde{\pi} \right\rangle \leq \left\| Q^\pi - \widehat{Q}^\pi \right\|_{L^2(v)} \left\| \tilde{\mu} \circ \tilde{\pi} \right\|_{L^2(v)}^* \leq \sqrt{\varepsilon_{\text{approx}}} \left\| \tilde{\mu} \circ \tilde{\pi} \right\|_{L^2(v)}^*,$$

where the last inequality is by our assumption. Further, by $v$-concentrability,

$$\left\| \tilde{\mu} \circ \tilde{\pi} \right\|_{L^2(v)}^* = \sqrt{\mathbb{E}_{s,a \sim v} \left( \frac{\tilde{\mu}(s) \tilde{\pi}_{s,a}}{v(s,a)} \right)^2} \leq \sqrt{C_v},$$

holds for $\tilde{\pi} \in \{\pi, \pi^+, \pi^\star\}, \tilde{\mu} \in \{\mu^\pi, \mu^\star\}$. Now, for such $\tilde{\mu}, \tilde{\pi}$, we have:

$$\left| \mathbb{E}_{s \sim \tilde{\mu}} \left\langle Q_s^\pi - \widehat{Q}_s^\pi, \pi_s - \tilde{\pi}_s \right\rangle \right| \leq \left| \mathbb{E}_{s \sim \tilde{\mu}} \left\langle Q_s^\pi - \widehat{Q}_s^\pi, \pi_s \right\rangle \right| + \left| \mathbb{E}_{s \sim \tilde{\mu}} \left\langle Q_s^\pi - \widehat{Q}_s^\pi, \tilde{\pi}_s \right\rangle \right| \leq 2\sqrt{C_v \varepsilon_{\text{approx}}},$$

therefore,

$$\left| \mathbb{E}_{s \sim \mu^\pi} \left\langle Q_s^\pi - \widehat{Q}_s^\pi, \pi_s - \pi_s^+ \right\rangle \right| \leq 2\sqrt{C_v \varepsilon_{\text{approx}}},$$

$$\left| \mathbb{E}_{s \sim \mu^\star} \left\langle Q_s^\pi - \widehat{Q}_s^\pi, \pi_s - \pi_s^\star \right\rangle \right| \leq 2\sqrt{C_v \varepsilon_{\text{approx}}}.$$

**Greedification.** Observe,

$$\mathbb{E}_{s \sim \mu^\pi} \left\langle Q_s^\pi, \pi_s - \pi_s^+ \right\rangle = \mathbb{E}_{s \sim \mu^\pi} \left\langle \widehat{Q}_s^\pi, \pi_s - \pi_s^+ \right\rangle + \mathbb{E}_{s \sim \mu^\pi} \left\langle Q_s^\pi - \widehat{Q}_s^\pi, \pi_s - \pi_s^+ \right\rangle$$

$$\geq \mathbb{E}_{s \sim \mu^\pi} \max_p \left\langle \widehat{Q}_s^\pi, \pi_s - p \right\rangle - \varepsilon_{\text{greedy}} - 2\sqrt{C_v \varepsilon_{\text{approx}}}$$

$$\implies \mathbb{E}_{s \sim \mu^\pi} \max_p \left\langle \widehat{Q}_s^\pi, \pi_s - p \right\rangle \leq \mathbb{E}_{s \sim \mu^\pi} \left\langle Q_s^\pi, \pi_s - \pi_s^+ \right\rangle + \varepsilon_{\text{greedy}} + 2\sqrt{C_v \varepsilon_{\text{approx}}}.$$

Therefore, by Lemma 14 (value difference),

$$
\begin{aligned}
\frac{1}{H}\left(V(\pi) - V^\star\right) &= \mathbb{E}_{s\sim\mu^\star}\left\langle Q_s^\pi, \pi_s - \pi_s^\star\right\rangle \\
&= \mathbb{E}_{s\sim\mu^\star}\left\langle \widehat{Q}_s^\pi, \pi_s - \pi_s^\star\right\rangle + \mathbb{E}_{s\sim\mu^\star}\left\langle Q_s^\pi - \widehat{Q}_s^\pi, \pi_s - \pi_s^\star\right\rangle \\
&\le \mathbb{E}_{s\sim\mu^\star}\max_{p\in\Delta(\mathcal{A})}\left\langle \widehat{Q}_s^\pi, \pi_s - p\right\rangle + 2\sqrt{C_\nu\varepsilon_{\text{approx}}} \\
&\le \left\|\frac{\mu^\star}{\mu^\pi}\right\|_\infty \mathbb{E}_{s\sim\mu^\pi}\max_{p\in\Delta(\mathcal{A})}\left\langle \widehat{Q}_s^\pi, \pi_s - p\right\rangle + 2\sqrt{C_\nu\varepsilon_{\text{approx}}} \\
&\le \left\|\frac{\mu^\star}{\mu^\pi}\right\|_\infty \mathbb{E}_{s\sim\mu^\pi}\left\langle Q_s^\pi, \pi_s - \pi_s^+\right\rangle + \left\|\frac{\mu^\star}{\mu^\pi}\right\|_\infty\left(\varepsilon_{\text{greedy}} + 2\sqrt{C_\nu\varepsilon_{\text{approx}}}\right) + 2\sqrt{C_\nu\varepsilon_{\text{approx}}} \\
&\le \left\|\frac{\mu^\star}{\mu^\pi}\right\|_\infty \mathbb{E}_{s\sim\mu^\pi}\left\langle Q_s^\pi, \pi_s - \pi_s^+\right\rangle + \left\|\frac{\mu^\star}{\mu^\pi}\right\|_\infty\left(\varepsilon_{\text{greedy}} + 4\sqrt{C_\nu\varepsilon_{\text{approx}}}\right) \\
&= \frac{1}{H}\left\|\frac{\mu^\star}{\mu^\pi}\right\|_\infty\left\langle \nabla V(\pi), \pi - \pi^+\right\rangle + \left\|\frac{\mu^\star}{\mu^\pi}\right\|_\infty\left(\varepsilon_{\text{greedy}} + 4\sqrt{C_\nu\varepsilon_{\text{approx}}}\right) \\
&\le \frac{1}{H}\left\|\frac{\mu^\star}{\mu^\pi}\right\|_\infty\max_{\tilde{\pi}\in\Pi}\left\langle \nabla V(\pi), \pi - \tilde{\pi}\right\rangle + \left\|\frac{\mu^\star}{\mu^\pi}\right\|_\infty\left(\varepsilon_{\text{greedy}} + 4\sqrt{C_\nu\varepsilon_{\text{approx}}}\right),
\end{aligned}
$$

which completes the proof after multiplying by $H$. $\qquad\square$

**Lemma 6** (Log-linear dual closure $\implies$ VGD). *Let $\left\{\phi_{s,a}\right\}_{s\in\mathcal{S},a\in\mathcal{A}} \subseteq \mathbb{R}^d$ be state-action feature vectors, and let $\Pi$ be the log-linear policy class $\Pi = \left\{\pi(\theta) \mid \theta \in \mathbb{R}^d\right\}$, where*

$$
\pi_{s,a}(\theta) := \frac{\exp(\phi_{s,a}^\top\theta)}{\sum_{a'\in\mathcal{A}}\exp(\phi_{s,a'}^\top\theta)}.
$$

*Assume further that for all $\pi \in \Pi$ it holds that*

$$
\min_w \mathbb{E}_{s,a\sim(\mu^\pi\circ\pi)}\left[\left(w^\top\phi_{s,a} - Q_{s,a}^\pi\right)^2\right] \le \varepsilon_{\text{approx}},
$$

*and,*

$$
\left\|\frac{\mu^\star}{\mu^\pi}\right\|_\infty \le \nu_\star,
$$

*and,*

$$
\mathbb{E}_{s,a\sim(\mu^\pi\circ\pi)}\left[\left(\frac{h_{s,a}^\pi}{\mu^k(s)\pi_{s,a}}\right)^2\right] \le C_\nu,
$$

*where $h^\pi$ represents $\tilde{\mu}\circ\tilde{\pi}$ for all $\tilde{\pi}\in\Pi, \tilde{\mu}\in\{\mu^\pi,\mu^\star\}$, and we denote $\pi^\star = \arg\min_{\pi\in\Delta(\mathcal{A})^\mathcal{S}} V(\pi), \mu^\star = \mu^{\pi^\star}$. Then $\Pi$ satisfies $(\nu_\star, 5\nu_\star H\sqrt{C_\nu\varepsilon_{\text{approx}}}; V^\star)$-VGD (Definition 7).*

*Proof.* Let $\pi \in \Pi$, and denote $\widehat{Q}_{s,a}^\pi = \phi_{s,a}^\top w_\star^\pi$ where

$$
w_\star^\pi := \arg\min_w \mathbb{E}_{s,a\sim(\mu^\pi\circ\pi)}\left[\left(w^\top\phi_{s,a} - Q_{s,a}^\pi\right)^2\right].
$$

By Lemma 7, the policy $\pi^+ := \pi(\theta^+) \in \Pi$ defined by $\theta^+ := (\log(d)/\varepsilon_{\text{greedy}})w_\star^\pi$ satisfies

$$
\forall s : \left\langle \widehat{Q}_s^\pi, \pi_s^+\right\rangle \le \min_a \widehat{Q}_{s,a}^\pi + \varepsilon_{\text{greedy}}.
$$

Now, by the above and our assumptions, we are in the position to apply Lemma 5, which immediately implies the desired for $\varepsilon_{\text{greedy}} = \sqrt{C_\nu\varepsilon_{\text{approx}}}$. $\qquad\square$

**Lemma 7** ((McSherry & Talwar, 2007; Epasto et al., 2020)). *Let $x_1, \ldots, x_d \in \mathbb{R}$. Then if $\tau \geq (\log d)/\delta$, it holds that*

$$\frac{\sum_i e^{-\tau x_i} x_i}{\sum_i e^{-\tau x_i}} \leq \min_i x_i + \delta.$$

*Proof.* We have

$$\frac{\sum_i e^{-\tau x_i} x_i}{\sum_i e^{-\tau x_i}} - \min_i x_i = \max_i \{-x_i\} - \frac{\sum_i e^{-\tau x_i}(-x_i)}{\sum_i e^{-\tau x_i}}$$

The result now follows from the original statement, which says that for any $z_1, \ldots, z_n \in \mathbb{R}$,

$$\max_i z_i - \frac{\sum_i e^{\tau z_i} z_i}{\sum_i e^{\tau z_i}} \leq \delta. \quad \square$$

Next, we provide a proof for closure conditions of Alfano et al. (2023) in the case of a regularizer with a bounded Bregman divergence, which simplifies some technical issues and is sufficient for the Euclidean case. The implication can be shown to hold more generally subject to some additional regularity conditions on the policy class. We note that such a general version of the lemma would in particular imply Lemma 6, thus rendering the above proof redundant. However, we opted for an independent proof of Lemma 6 to avoid the additional regularity assumptions.

**Lemma 8** (Generic dual closure $\implies$ VGD). *Let $\Pi \subset \Delta(\mathcal{A})^{\mathcal{S}}$ be a policy class, and $R \colon \mathbb{R}^A \to \mathbb{R}$ be an action regularizer. For any policy $\pi$ let $\eta > 0$ be a chosen step size and $\nu$ be a chosen state-action probability measure. Define*

$$f^+ := f^+(\pi, \eta) := \arg\min_{f \in \mathcal{F}} \left\| f - \left(\eta^{-1} \nabla R(\pi) - Q^\pi\right) \right\|^2_{L^2(\nu)}$$

$$\pi^+ := \pi^+(\pi, \eta) := P_R(\eta f^+),$$

*where $P_R(\eta f)_s := \Pi^R_{\Delta(\mathcal{A})}(\nabla R^*(\eta f_s))$. Assume that:*

$$\left\| f^+ - \left(\eta^{-1} \nabla R(\pi) - Q^\pi\right) \right\|^2_{L^2(\nu)} \leq \varepsilon_{\text{approx}}, \tag{A1}$$

*and for $\tilde{\pi} \in \{\pi, \pi^+, \pi^\star\}, \tilde{\mu} \in \{\mu^\pi, \mu^\star\}$:*

$$\mathbb{E}_{s,a \sim \nu} \left[ \left( \frac{\tilde{\mu}(s) \tilde{\pi}_{s,a}}{\nu(s,a)} \right)^2 \right] \leq C_\nu, \tag{A2}$$

*and finally,*

$$\sup_s \frac{\mu^\star(s)}{\mu^\pi(s)} \leq \nu_\star. \tag{A3}$$

*Then, if $R$ has a bounded Bregman divergence, $B \geq \max_{p,q \in \Delta(\mathcal{A})} B_R(p,q)$, and the above holds for any $\eta$, it holds that $\Pi$ satifies $\left(\nu_\star, 5H\nu_\star\sqrt{C_\nu \varepsilon_{\text{approx}}}; V^\star\right)$-VGD (Definition 7).*

*Proof.* Fix $\pi \in \Pi$, and define

$$\widehat{Q}^\pi := \eta^{-1} \nabla R(\pi) - f^+$$
$$\implies f^+ = \eta^{-1} \nabla R(\pi) - \widehat{Q}^\pi,$$

which implies that:

$$\forall s, \ \pi^+_s = \arg\min_{p \in \Delta(\mathcal{A})} \left\langle \widehat{Q}^\pi_s, p \right\rangle + \frac{1}{\eta} B_R(p, \pi_s)$$

We have:

$$\left\| Q^\pi - \widehat{Q}^\pi \right\|^2_{L^2(\nu)} = \left\| f^+ - \left( \eta^{-1} \nabla R(\pi) - Q^\pi \right) \right\|^2_{L^2(\nu)} \leq \varepsilon_{\text{approx}},$$

and for $\eta = B/\varepsilon_{\text{greedy}}$, by Lemma 9:

$$\forall s, \ \left\langle \widehat{Q}^\pi_s, \pi^+_s \right\rangle \leq \min_a \widehat{Q}^\pi_{s,a} + \varepsilon_{\text{greedy}}.$$

Choosing $\varepsilon_{\text{greedy}} = \sqrt{C_\nu \varepsilon_{\text{approx}}}$, the result follows by Lemma 5. $\qquad \square$

**Lemma 9.** *Let $\epsilon > 0$, $R \colon \mathbb{R}^A \to \mathbb{R}$ be a convex regularizer with bounded Bregman divergence $B \geq \max_{p,q \in \Delta(\mathcal{A})} B_R(p, q)$, and $g \in \mathbb{R}^A$ be a linear objective, with $a^\star = \arg\min_a g_a$. Then, for any $x \in \Delta(A)$, for $\eta \geq B/\epsilon$, we have:*

$$x^+ = \arg\min_{z \in \Delta(A)} \left\{ \langle g, z \rangle + \frac{1}{\eta} B_R(z, x) \right\} \implies g(x^+) \leq g_{a^\star} + \epsilon.$$

*Proof.* By optimality of $x^+$:

$$\begin{aligned} g(x^+) &\leq g(e_{a^\star}) + \frac{1}{\eta} B_R(e_{a^\star}, x) - \frac{1}{\eta} B_R(x^+, x) \\ &\leq g_{a^\star} + B/\eta \\ &= g_{a^\star} + \epsilon, \end{aligned}$$

and the result follows. $\qquad \square$

### A.2. Closure without convexity

In this section, we explain how the approximate closure conditions of Alfano et al. (2023) eliminate the need for convexity of $\Pi$ in our analysis. Roughly speaking, closure conditions imply approximate optimality conditions hold for the PMD iterates w.r.t. the complete policy class. And, in our analysis, we obtain guarantees w.r.t. the policy class the PMD iterates satisfy optimality conditions with respect to, regardless of actual policy class the algorithm operates over. To make the argument formal, we consider the following assumption, which characterizes the behavior of the algorithm in relation to an "ambient" policy class $\widetilde{\Pi}$.

**Assumption 4** (PMD w.r.t. ambient $\widetilde{\Pi}$). For $\widetilde{\Pi}$ a policy class, and $\pi^1, \ldots, \pi^{K+1}$ is a sequence of policies, it holds that:

1. $\widetilde{\Pi}$ is convex.

2. $\widetilde{\Pi}$ satisfies $(C_\star, \varepsilon_{\text{vgd}}; v^\star)$-VGD on the iterates $\pi^1, \ldots, \pi^{K+1}$:

$$V(\pi^k) - v^\star \leq C_\star \max_{\tilde\pi \in \widetilde{\Pi}} \left\langle \nabla V(\pi^k), \pi^k - \tilde\pi \right\rangle + \varepsilon_{\text{vgd}}.$$

3. $\pi^1, \ldots, \pi^{K+1}$ satisfy PMD approximate optimality conditions w.r.t. $\widetilde{\Pi}'$:

$$\forall \pi \in \widetilde{\Pi}', \left\langle \nabla \phi_k(\pi^{k+1}), \pi - \pi^{k+1} \right\rangle \geq -\varepsilon_{\text{act}},$$

where $\phi_k(\pi) := \mathbb{E}_{s \sim \mu^k} \left[ \langle Q^k_s, \pi_s \rangle + \frac{1}{\eta} B_R(\pi_s, \pi^k_s) \right]$.

We note that a PMD algorithm does not necessarily need access to $\widetilde{\Pi}$ to satisfy Assumption 4. In particular, it may be that the algorithm operates over non-convex $\Pi$, but satisfies Assumption 4 with $\widetilde{\Pi} = \widetilde{\Pi}' = \Delta(\mathcal{A})^{\mathcal{S}}$. Next, we restate our guarantees for the Euclidean case reframed in the context of Assumption 4, and then proceed to demonstrate assumptions of Alfano et al. (2023) imply Assumption 4.

***Theorem*** (Restatement of Theorem 1; Euclidean case). Let $\widetilde{\Pi} \subset \Delta(\mathcal{A})^{\mathcal{S}}$ be a policy class and suppose $\pi^1, \pi^2, \ldots, \pi^{K+1}$ is a sequence of policies for which Assumption 4 holds with $\widetilde{\Pi}' = \widetilde{\Pi}^{\varepsilon_{\text{expl}}}$, $R(p) = \frac{1}{2} \|p\|^2_2$, and $\eta, \varepsilon_{\text{expl}}$ properly tuned. Then, it holds that:

$$V(\pi^K) - v^\star = O\left( \frac{C_\star^2 A^{3/2} H^3}{K^{2/3}} + \left( C_\star H + A H^2 K^{1/6} \right) \sqrt{\varepsilon_{\text{act}}} + \varepsilon_{\text{vgd}} \right)$$

To establish the next lemma, we interpret the colure conditions of Alfano et al. (2023) as perfect closure, where $\varepsilon_{\text{approx}}$ bounds the actor error, rather than relating to expressivity of the dual policy parametrization.

**Lemma 10.** *Suppose that for all $k \in [K]$,*

$$\left\| f^{k+1} - \left( \eta^{-1} \nabla R(\pi^k) - Q^k \right) \right\|_{L^2(v^k)}^2 \leq \varepsilon_{\text{approx}}, \tag{A1}$$

*and $\pi^{k+1} = P_R(\eta f^{k+1})$ where $P_R(\eta f)_s := \Pi^R_{\Delta(\mathcal{A})}(\nabla R^*(\eta f_s))$. Suppose further that for all $k$,*

$$\sup_s \frac{\mu^\star(s)}{\mu^k(s)} \leq \nu_\star. \tag{A3}$$

*Then, with the choice of $v^k = \mu^k \circ u$, i.e., $s, a \sim v^k \implies s \sim \mu^k, a \sim \text{Unif}(\mathcal{A})$, we have that Assumption 4 is satisfied with $\widetilde{\Pi} = \widetilde{\Pi}' = \Delta(\mathcal{A})^{\mathcal{S}}$, $v^\star = V^\star$, $C_\star = \nu_\star$, $\varepsilon_{\text{vgd}} = 0$, and $\varepsilon_{\text{act}} \leq 2\sqrt{A\varepsilon_{\text{approx}}}$.*

*Proof.* Let $\zeta^{k+1} := f^{k+1} - \left( \eta^{-1} \nabla R(\pi^k) - Q^k \right)$. Then $\left\| \zeta^{k+1} \right\|_{L^2(v^k)}^2 \leq \varepsilon_{\text{approx}}$, and

$$f^{k+1} = \eta^{-1} \nabla R(\pi^k) - \left( Q^k + \zeta^{k+1} \right).$$

Now by definition of $\pi^{k+1}$,

$$\pi_s^{k+1} = \underset{\pi_s \in \Delta(\mathcal{A})}{\arg \min} \left\langle Q_s^k + \zeta_s^{k+1}, \pi_s \right\rangle + \frac{1}{\eta} B_R(\pi_s, \pi_s^k)$$

hence, by optimality conditions, for any $\pi \in \Delta(\mathcal{A})^{\mathcal{S}}$:

$$\left\langle Q_s^k + \zeta_s^{k+1} + \frac{1}{\eta} \left( \nabla R(\pi_s^{k+1}) - \nabla R(\pi_s^k) \right), \pi_s - \pi_s^{k+1} \right\rangle \geq 0$$

$$\iff \left\langle Q_s^k + \frac{1}{\eta} \left( \nabla R(\pi_s^{k+1}) - \nabla R(\pi_s^k) \right), \pi_s - \pi_s^{k+1} \right\rangle \geq \left\langle \zeta_s^{k+1}, \pi_s^{k+1} - \pi_s \right\rangle$$

Now, note that

$$\begin{aligned}
\mathbb{E}_{s \sim \mu^k} \left\langle \zeta_s^{k+1}, \pi_s^{k+1} - \pi_s \right\rangle &= \left\langle \mu^k \circ \zeta^{k+1}, \pi^{k+1} - \pi \right\rangle \\
&\leq \left\| \mu^k \circ \zeta^{k+1} \right\|_{L^2(\mu^k),2}^* \left\| \pi^{k+1} - \pi \right\|_{L^2(\mu^k),2} \\
&\leq 2\sqrt{\mathbb{E}_{s \sim \mu^k} \left\| \zeta_s^{k+1} \right\|_2^2}
\end{aligned}$$

Further, by the choice of $v^k = \mu^k \circ u$,

$$\mathbb{E}_{s \sim \mu^k} \left\| \zeta_s^{k+1} \right\|_2^2 = \mathbb{E}_{s \sim \mu^k} \left[ \sum_{a \in \mathcal{A}} \left( \zeta_{s,a}^{k+1} \right)^2 \right] = A \mathbb{E}_{s \sim \mu^k} \left[ \sum_{a \in \mathcal{A}} \frac{1}{A} \left( \zeta_{s,a}^{k+1} \right)^2 \right] = A \left\| \zeta^{k+1} \right\|_{L^2(v^k)}^2.$$

Therefore, for all $\pi \in \Delta(\mathcal{A})^{\mathcal{S}}$:

$$\mathbb{E}_{s \sim \mu^k} \left\langle Q_s^k + \frac{1}{\eta} \left( \nabla R(\pi_s^{k+1}) - \nabla R(\pi_s^k) \right), \pi_s - \pi_s^{k+1} \right\rangle \geq -\varepsilon_{\text{act}}$$

with $\varepsilon_{\text{act}} \leq 2\sqrt{A\varepsilon_{\text{approx}}}$. Finally, the complete class satisfies $(v^\star, 0)$-VGD on the $\pi^k$ iterates by Lemma 16, with $v^\star$ in place of $H\nu_0$ owed to our assumption (A3). □

Finally, we note that we could have traded the dependence on the action set with an additional concentrability assumption.

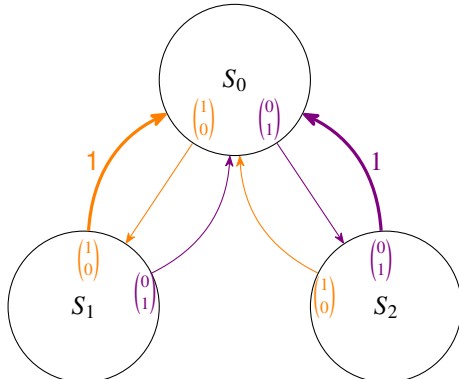

Figure 1: A simple MDP with a convex value landscape. Each action represented by a (feature-vector, edge) pair leads deterministically to the state at the other end of the edge. The two outer bold edges labeled 1 inflict a cost of 1, the others have cost 0.

### A.3. VGD does not imply closure

In this section, we present a sinple example where the VGD condition holds but closure does not, and as a result existing analyses fail to establish convergence of PMD. We note that the fact that VGD does not imply closure is immediate, as closure implies realizability but VGD does not. We go further here to show that the bounds of prior works may indeed become vacuous in setups where VGD holds and closure does not. We consider the MDP depicted in Figure 1 with the log-linear policy class $\Pi$ induced by the state-action feature vectors shown in the diagram.

For simplicity we assume there are no statistical errors in the execution of the algorithm ($\varepsilon_{\text{stat}} = 0$). In this example the value landscape is convex (in state-action space) over $\Pi$, and thus $\Pi$ is $(1, 0)$-VGD and convergence of PMD follows by our main theorem:

$$V(\pi^K) - \min_{\pi^\star \in \Pi} V(\pi^\star) \xrightarrow[K \to \infty]{} 0.$$

At the same time, results based on closure imply convergence to an error floor that is larger than $H$. For instance, by Theorem 1 of Yuan et al. (2023) establishes that:

$$V(\pi^K) - \min_{\pi^\star \in \Pi} V(\pi^\star) \lesssim 2H \left(1 - \frac{1}{\nu_0}\right)^K + 2H\nu_0 \sqrt{AC_0 \varepsilon_{\text{bias}}}, \tag{9}$$

meaning:

$$V(\pi^K) - \min_{\pi^\star \in \Pi} V(\pi^\star) \xrightarrow[K \to \infty]{} 2H\nu_0 \sqrt{AC_0 \varepsilon_{\text{bias}}} \geq 10H,$$

where $\varepsilon_{\text{bias}} = \Omega(1)$, $\nu_0 := H \left\| \frac{\mu^\star}{\rho_0} \right\|_\infty$, and $C_0$ is a certain concentrability coefficient larger than 1. Here, both the transfer error and approximation error are $\Omega(\varepsilon_{\text{bias}})$. A rigorous analysis is given below in Appendix A.3.1. Recent papers such as Alfano et al. (2023); Xiong et al. (2024) accommodate more general policy parameterizations but still include the log-linear setup as a special case (see discussion in Alfano et al., 2023 and Appendix F). The error floor in their results is also larger than $H$ for the example in question for exactly the same reasons; their results depend on the approximation error, which for this example as mentioned behaves the same as the transfer error.

Finally, we note that the example is not realizable and the discussion focuses on best-in class convergence as the objective. If we were to look for convergence w.r.t. the true optimal policy, our Theorem 1 establishes convergence to an error floor of $V(\Pi^\star) - V^\star \approx H/2$, while closure based analyses suffer from the same $\geq H$ error floor. In all that follows, we focus on the transfer error $\varepsilon_{\text{bias}}$; the argument for the approximation error is the same.

### A.3.1. ANALYSIS

We denote the actions:

$$u := \begin{pmatrix} 1 \\ 0 \end{pmatrix}, \quad b := \begin{pmatrix} 0 \\ 1 \end{pmatrix},$$

and the state-action features, for all $s$:

$$\phi_{s,1} := \begin{pmatrix} 1 \\ 0 \end{pmatrix}, \quad \phi_{s,2} := \begin{pmatrix} 0 \\ 1 \end{pmatrix}, \quad \phi_s := (\phi_{s,1}, \phi_{s,2}) = \left( \begin{pmatrix} 1 \\ 0 \end{pmatrix} \begin{pmatrix} 0 \\ 1 \end{pmatrix} \right) \in \mathbb{R}^{2 \times 2}.$$

In favor of conciseness, we will let

$$\phi_{i,\cdot} := \phi_{S_i,\cdot}.$$

For $\theta \in \mathbb{R}^2$, we denote the log-linear policy $\pi_s^\theta := \sigma(\phi_s^\top \theta)$, where $\sigma$ is the softmax function:

$$\sigma(u)_i := \frac{e^{u_i}}{\sum_j e^{u_j}}.$$

This gives rise to the log-linear policy class:

$$\Pi := \left\{ \pi^\theta \mid \theta \in \mathbb{R}^2 \right\}.$$

Since such a policy $\pi^\theta$ in this MDP must select actions independent of the state, we let $\alpha$ denote the probability it chooses $u$ and $1 - \alpha$ the probability it chooses $b$; $\alpha := \pi_{s,u}^\theta \implies 1 - \alpha = \pi_{s,b}^\theta$. Now, denote $V_i^\alpha := V_{S_i}\left(\pi^\theta\right), Q_{i,\cdot}^\alpha := Q_{S_i,\cdot}^{\pi^\theta}$, and observe that by direct computation:

$$V_0(\alpha) = \frac{\gamma}{(1 - \gamma)(1 + \gamma)} \left( \alpha^2 + (1 - \alpha)^2 \right) =: \widetilde{H} \left( \alpha^2 + (1 - \alpha)^2 \right)$$

$$V_1(\alpha) = \alpha + \gamma \widetilde{H} \left( \alpha^2 + (1 - \alpha)^2 \right)$$

$$V_2(\alpha) = (1 - \alpha) + \gamma \widetilde{H} \left( \alpha^2 + (1 - \alpha)^2 \right).$$

and,

$$\begin{aligned} Q_{0,u}^\alpha &= \gamma V_1(\alpha), & Q_{0,b}^\alpha &= \gamma V_2(\alpha); \\ Q_{1,u}^\alpha &= 1 + \gamma V_0(\alpha), & Q_{1,b}^\alpha &= \gamma V_0(\alpha); \\ Q_{2,u}^\alpha &= \gamma V_0(\alpha), & Q_{2,b}^\alpha &= 1 + \gamma V_0(\alpha). \end{aligned}$$

Let $\rho_0(S0) = 1 - p, \rho_0(S1) = \rho_0(S2) = p/2$ for some $p \in [0, 1)$. Then

$$V(\alpha) = (1 - p + \gamma p) \widetilde{H} \left( \alpha^2 + (1 - \alpha)^2 \right) + p/2.$$

**The VGD condition holds.** It is not hard to verify the value function is convex (in state-action space) over this policy class. Indeed, we have

$$\left\langle \nabla_{\pi^\theta} V(\pi^\theta), \pi^{\tilde\theta} - \pi^\theta \right\rangle = \frac{\partial V^\alpha}{\partial \alpha} \left( \tilde\alpha - \alpha \right),$$

and therefore convexity of $V^\alpha$ w.r.t. $\alpha$ implies convexity in the direct parametrization over $\Pi$. Hence in particular, $\Pi$ is $(1, 0)$-VGD w.r.t. the MDP in question. Thus, convergence of PMD follows by Theorem 1, which in this case guarantees the sub-optimality of $\pi^K$ tends to 0 as $K$ grows (since there is no error floor).

**Closure does not hold, and the error floor in closure based analyses is $\geq H = \frac{1}{1-\gamma}$.** Let $\mu^\alpha := \mu^{\pi^\theta}$, then

$$\mu^\alpha(S_0) = \frac{(1 - \gamma)(1 - p) + \gamma}{1 + \gamma} = \frac{1 - p + \gamma H}{(1 + \gamma)H}$$

$$\mu^\alpha(S_1) = (1 - \gamma)p + \gamma \alpha \mu^\alpha(S_0)$$

$$\mu^\alpha(S_2) = (1 - \gamma)p + \gamma(1 - \alpha)\mu^\alpha(S_0).$$

It is immediate that the optimal in-class policy is given by $\theta^\star := (1, 1), \alpha^\star = 1/2$, and satisfies,

$$\mu^\star(S_0) = \frac{1 - p + \gamma H}{(1 + \gamma)H}, \quad \mu^\star(S_1) = \frac{H + p - 1}{2(1 + \gamma)H}, \quad \mu^\star(S_2) = \frac{H + p - 1}{2(1 + \gamma)H}.$$

Now suppose that $\gamma \geq 0.99$ and $p \leq 1/100$, then by direct computation,

$$\mu^\star(S_0) \approx \frac{1}{2}, \quad \mu^\star(S_1) \approx \frac{1}{4}, \quad \mu^\star(S_2) \approx \frac{1}{4},$$

where the approximation is correct up to error of $1/100$. Recall that for a policy $\pi^{(k)}$, in the NPG update step ([Agarwal et al., 2021](); [Yuan et al., 2023]())

$$w_\star^{(k)} := \arg\min_w \mathbb{E}_{s \sim \mu^k, a \sim \pi_s^k} \left[ \left( \phi_{s,a}^\top w - Q_{s,a}^k \right)^2 \right]$$

Meanwhile, by definition

$$
\begin{aligned}
\epsilon_{\text{bias}} &\geq \mathbb{E}_{s \sim \mu^\star, a \sim \text{Unif}(\mathcal{A})} \left[ \left( \phi_{s,a}^\top w_\star^{(k)} - Q_{s,a}^k \right)^2 \right] \\
&\geq \frac{1}{2} \arg\min_{w_1} \mathbb{E}_{s \sim \mu^\star} \left[ \left( w_1 - Q_{s,u}^k \right)^2 \right] \\
&\approx \frac{1}{2} \arg\min_{w_1} \left\{ \frac{1}{2} (w_1 - \gamma V_1(\alpha))^2 + \frac{1}{4} (w_1 - 1 - \gamma V_0(\alpha))^2 + \frac{1}{4} (w_1 - \gamma V_0(\alpha))^2 \right\} \\
&\geq \frac{1}{8} \arg\min_{w_1} \left\{ (w_1 - 1 - \gamma V_0(\alpha))^2 + (w_1 - \gamma V_0(\alpha))^2 \right\} \\
&= \frac{1}{32}
\end{aligned}
$$

Now the bias term in Equation (9) is at least as large as

$$H \left\| \frac{\mu^\star}{\rho_0} \right\|_\infty \sqrt{\epsilon_{\text{bias}}} \gtrsim H \frac{1}{p} \sqrt{\frac{1}{32}} \geq 10H.$$

### A.4. Additional Remarks

In this section we include several additional points for consideration regarding closure and VGD conditions.

**On-policy PMD is prone to local optima.** The necessity of some structural assumption (whether VGD or closure) is motivated in the introduction by the fact that policy gradient methods over non-complete policy classes $\Pi \neq \Delta(\mathcal{A})^S$ are prone to local optima ([Bhandari & Russo, 2024]()). While PMD and vanilla policy gradients are not the same algorithm, the example given in [Bhandari & Russo (2024)]() (Example 1) also applies to PMD with Euclidean regularization, as we explain next. A vanilla policy gradient update in the direct parametrization case is equivalent to:

$$\pi^{k+1} = \arg\min_{\pi \in \Pi} \left[ \mathbb{E}_{s \sim \mu^k} \left[ \langle Q_s^k, \pi_s \rangle \right] + \frac{1}{2\eta} \|\pi - \pi^k\|_2^2 \right],$$

which is an "unweighted regularization" version of Euclidean PMD. While this is equivalent to PMD for $\Pi = \Delta(\mathcal{A})^S$ (in the error free case), it is indeed not equivalent in general. However, Example 1 of [Bhandari & Russo (2024)]() indeed also applies to Euclidean PMD because the policy class in question contains only policies $\pi$ such that $\pi_{s,a} = \pi_{s',a}$ for all $s, s', a$. Hence, for any two policies $\pi, \pi^k \in \Pi$, $\|\pi_s - \pi_s^k\|_2^2 = \|\pi_{s'} - \pi_{s'}^k\|_2^2$ for all $s, s'$, and $\|\pi - \pi_k\|_2^2 = S \mathbb{E}_{s \sim \mu^k} \|\pi_s - \pi_s^k\|_2^2 = 2\mathbb{E}_{s \sim \mu^k} \|\pi_s - \pi_s^k\|_2^2$. Thus, for the example in question the two algorithms are equivalent up to scaling of the step-size by a constant factor.

**Closure conditions in practice.**   Closure conditions (that are based on bounded approximation error) roughly stipulate the policy class is closed to a soft policy improvement step. This has a flavor that is similar to Bellman completeness (Munos & Szepesvári, 2008; Chen & Jiang, 2019; Zanette et al., 2020; Zanette, 2023), a property of a $Q$-function class that says the class is closed to a Bellman backup step. Bellman completeness is widely considered too strong a condition to hold in practice, the reasoning being that increasing capacity of a function class that violates completeness inadvertently introduces new functions for which completeness needs to be satisfied. Therefore, an increase in capacity may actually cause completeness to be further violated. The same can be argued for closure conditions, with one difference being that the complete policy class $\Delta(\mathcal{A})^S$ is naturally closed to any policy improvement step. However, in a large scale environment setting, the complete policy class is typically many orders of magnitude too large to be well approximated by realistically sized neural network architectures (at least at the present time).

**PMD and VGD from the optimization perspective.**   Standard arguments from optimization literature are insufficient to establish convergence of PMD with the VGD condition. First, PMD is not an algorithm that has (prior to our work) a formulation within a purely optimization-based framework. Second, convergence in a smooth non-convex setting typically scales with the distance to the optimal solution, measured by the norm induced by smoothness of the objective. Prior works that establish convergence of gradient descent type methods (though not of PMD; e.g., Agarwal et al., 2021; Bhandari & Russo, 2024; Xiao, 2022) exploit smoothness of the value function w.r.t. the Euclidean norm (established in Agarwal et al., 2021), and as a result obtain bounds that scale with the cardinality of the state-space.

**Divergence of Policy Iteration.**   Our setup with the VGD condition is general enough to accommodate examples where the policy iteration algorithm does not converge (the same example we discuss in Appendix A.3 demonstrates this). Here, since the policy class is non-complete, the policy improvement step is performed over the current policy occupancy measure (see Bhandari & Russo, 2024 who introduce this natural adaptation). Arguably, it should not be expected that policy iteration converges for real world, large-scale problems, as it is a very "non-regularized" algorithm from an optimization perspective. At the same time, in setups where closure conditions based on bounded approximation error hold, in particular, closure to policy improvement as studied in Bhandari & Russo (2024), the policy iteration algorithm converges at a linear rate. Thus it is not immediately clear why should we employ more sophisticated algorithms such as PMD in such settings.

**Convergence beyond the VGD condition.**   Using our framework, it can be shown that PMD converges to a stationary point regardless of any VGD condition; see Appendix E.3.

## B. Deferred Discussions

### B.1. Assumption on the critic error

Our results can be easily adapted to the (generally weaker) assumption that

$$\mathbb{E}_{s \sim \mu^\pi} \left\| \widehat{Q}_s^\pi - Q_s^\pi \right\|_2 \leq \varepsilon_{\text{crit}}.$$

(In which case the bounds would depend on $\varepsilon_{\text{crit}}$ rather than $\sqrt{\varepsilon_{\text{crit}}}$.) Assumption 2 in its current form simplifies presentation, since it allows working with the weighted $L^2$ norm for both smoothness and errors in the gradient approximation. Also noteworthy, when working with the negative entropy regularizer, approximation w.r.t. the $\|\cdot\|_\infty$ norm would suffice. Since the statistical errors are not the focus of this work, we make these concessions in favor of a more streamlined and clear presentation.

### B.2. Local smoothness of the value function requires greedy exploration

In this section we discuss why the dependence on $\epsilon$ in the bound of Lemma 2 cannot be improved in general. We consider the MDP in Figure 2, for which we can show Lemma 2 has tight dependence on the $\epsilon$-exploration parameter. Let $p \in (0, 1/2)$ and $0 < \epsilon < p$. Define:

$$\pi := \pi^\epsilon, \quad \pi_{1,0} = 1, \pi_{1,0} = 0,$$
$$\tilde{\pi} := \pi^p, \quad \tilde{\pi}_{1,0} = 0, \tilde{\pi}_{1,0} = 1.$$

**Idea.**   Think of $\epsilon$ as much smaller than $p$. When measuring distance with the local norm $\|\tilde{\pi} - \pi\|_{L^2(\mu^\pi),1}$, the large difference $\|\tilde{\pi}_1 - \pi_1\|_1^2$ gets little weight: $\mu^\pi(S_1) \approx \epsilon$. Meanwhile, the error of the linear approximation at $\pi$ behaves like

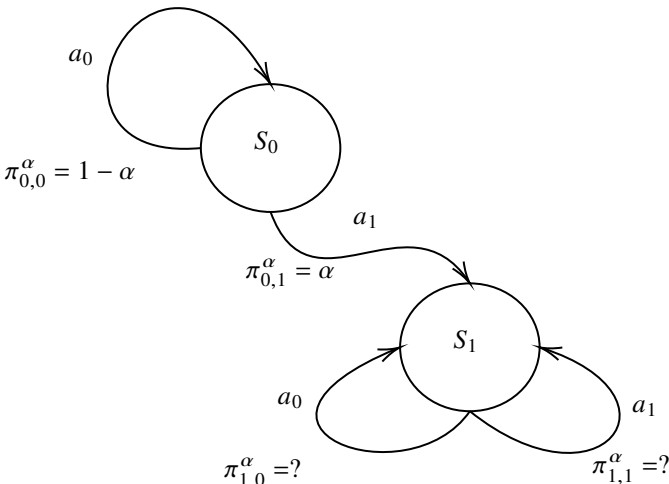

Figure 2: A two state deterministic MDP, with $\rho_0(S_0) = 1$. Each edge is labeled with an action ($a \in \{a_0, a_1\}$) that takes the agent to the state at the other end. A policy $\pi^\alpha$, $\alpha \in [0, 1]$ takes actions in $S_0$ with the probabilities displayed in the diagram next to the relevant action. The probabilities $\pi^\alpha$ assigns to actions in $S_1$ denoted by ? are unrelated to $\alpha$ and left for later.

(see proof of Lemma 2 in Appendix D.2):

$$\left| \sum_s \mu^\pi(s) \sum_a (\tilde{\pi}_{sa} - \pi_{sa}) \left( \sum_{s'} \mu^\pi_{\mathbb{P}_{sa}}(s') \, \|\tilde{\pi}_{s'} - \pi_{s'}\|_1 \right) \right|,$$

where the weight assigned to $\|\tilde{\pi}_1 - \pi_1\|_1^2$ is approximately $(\tilde{\pi}_{0,1} - \pi_{0,1}) = p - \epsilon$. Hence, if $\epsilon = p^2$,

$$|V(\tilde{\pi}) - V(\pi) - \langle \nabla V(\pi), \tilde{\pi} - \pi \rangle| \approx p,$$
$$\|\tilde{\pi} - \pi\|_{L^2(\mu^\pi),1}^2 \approx p^2,$$

so

$$|V(\tilde{\pi}) - V(\pi) - \langle \nabla V(\pi), \tilde{\pi} - \pi \rangle| \gtrsim \frac{1}{\sqrt{\epsilon}} \, \|\tilde{\pi} - \pi\|_{L^2(\mu^\pi),1}^2 \, .$$

**Computations.** The term that is equal to the linearization error, up to constant factors, is the following:

$$\left| \sum_s \mu^\pi(s) \sum_a (\tilde{\pi}_{sa} - \pi_{sa}) \left( \sum_{s'} \mu^\pi_{\mathbb{P}_{sa}}(s') \left\langle Q^{\tilde{\pi}}_{s'}, \tilde{\pi}_{s'} - \pi_{s'} \right\rangle \right) \right|.$$

Assume $p > \epsilon$. By choosing a cost function $r(s, i) = i$ for $s \in \{S_0, S_1\}$, $i \in \{0, 1\}$ we have that for all $s$,

$$\left\langle Q^{\tilde{\pi}}_s, \tilde{\pi}_s - \pi_s \right\rangle = \Omega(\|\tilde{\pi}_s - \pi_s\|_1),$$

hence we focus on lower bounding

$$(*) := \left| \sum_s \mu^\pi(s) \sum_a (\tilde{\pi}_{sa} - \pi_{sa}) \left( \sum_{s'} \mu^\pi_{\mathbb{P}_{sa}}(s') \, \|\tilde{\pi}_{s'} - \pi_{s'}\|_1 \right) \right|.$$

By direct computation,

$$\mu^{\pi}(S_0) = \frac{1}{1 + \gamma\epsilon}, \quad \mu^{\pi}(S_1) = \frac{\gamma\epsilon}{(1 + \gamma\epsilon)(1 - \gamma)}$$

and

$$\|\tilde{\pi}_0 - \pi_0\|_1 = 2|p - \epsilon|, \quad \|\tilde{\pi}_1 - \pi_1\|_1 = 2.$$

Thus,

$$(\tilde{\pi}_{0,0} - \pi_{0,0}) \sum_{s'} \mu^{\pi}_{\mathbb{P}_{0,0}}(s') \|\tilde{\pi}_{s'} - \pi_{s'}\|_1 \approx (1 - \epsilon)(\epsilon - p)|p - \epsilon| + \epsilon \geq -p^2 + \epsilon$$

$$(\tilde{\pi}_{0,1} - \pi_{0,1}) \sum_{s'} \mu^{\pi}_{\mathbb{P}_{0,1}}(s') \|\tilde{\pi}_{s'} - \pi_{s'}\|_1 \approx p - \epsilon,$$

and further,

$$\left| \sum_a \left( \tilde{\pi}_{1,a} - \pi_{1,a} \right) \left( \sum_{s'} \mu^{\pi}_{\mathbb{P}_{1,a}}(s') \|\tilde{\pi}_{s'} - \pi_{s'}\|_1 \right) \right| = 0.$$

We obtain

$$(*) \gtrsim \mu^{\pi}(S_0) \left( p - p^2 \right) \approx p - p^2.$$

Meanwhile,

$$\|\tilde{\pi} - \pi\|^2_{L^2(\mu^{\pi}),1} = \frac{4(p - \epsilon)^2}{1 + \gamma\epsilon} + \frac{4\gamma\epsilon}{(1 + \gamma\epsilon)(1 - \gamma)} \approx (p - \epsilon)^2 + \epsilon.$$

Now,

$$\frac{|V(\tilde{\pi}) - V(\pi) - \langle \nabla V(\pi), \tilde{\pi} - \pi \rangle|}{\|\tilde{\pi} - \pi\|^2_{L^2(\mu^{\pi}),1}} \approx \frac{(*)}{\|\tilde{\pi} - \pi\|^2_{L^2(\mu^{\pi}),1}} \approx \frac{p - p^2}{(p - \epsilon)^2 + \epsilon}.$$

Now, for $\epsilon := p^2$, $p < 1/2$, we obtain

$$\frac{p - p^2}{(p - \epsilon)^2 + \epsilon} = \frac{p - p^2}{(p - p^2)^2 + p^2} \geq \frac{p}{4p^2} = \frac{1}{4p} = \frac{1}{4\sqrt{\epsilon}}.$$

## C. State-weighted state-action space: Basic Facts

Given a state probability measure $\mu \in \Delta(\mathcal{S})$, and an action space norm $\|\cdot\|_{\circ} : \mathbb{R}^A \to \mathbb{R}$, we define the induced state-action weighted $L^p$ norm $\|\cdot\|_{L^p(\mu),\circ} : \mathbb{R}^{SA} \to \mathbb{R}$:

$$\|u\|_{L^p(\mu),\circ} := \left( \mathbb{E}_{s\sim\mu} \|u_s\|^p_{\circ} \right)^{1/p}. \tag{10}$$

In addition, for $\mu \in \mathbb{R}^S, Q \in \mathbb{R}^{SA}$, we define the state to state-action element-wise product $\mu \circ Q \in \mathbb{R}^{SA}$:

$$(\mu \circ Q)_{s,a} := \mu(s)Q_{s,a}. \tag{11}$$

**Lemma 11.** *For any strictly positive measure $\mu \in \mathbb{R}^S_{++}$, the dual norm of $\|\cdot\|_{L^2(\mu),\circ}$ is given by*

$$\|z\|^*_{L^2(\mu),\circ} = \sqrt{\int \mu(s)^{-1} \left( \|z_s\|^*_{\circ} \right)^2 ds} \tag{12}$$

*Proof.* First denote

$$z_s^* := \arg\max_{u_s \in \mathbb{R}^A, \|u_s\|_\circ \le 1} \langle u_s, z_s \rangle$$

$$\implies \|z_s\|_\circ^* = \langle z_s^*, z_s \rangle, \text{ and } \|z_s^*\|_\circ = 1.$$

Now let $x \in \mathbb{R}^{SA}$ be defined by $x_s := \frac{\|z_s\|_\circ^*}{\mu(s)} z_s^*$, then

$$\langle x, z \rangle = \int \frac{\|z_s\|_\circ^*}{\mu(s)} \langle z_s^*, z_s \rangle \, \mathrm{d}s = \int \frac{1}{\mu(s)} \left( \|z_s\|_\circ^* \right)^2 \mathrm{d}s.$$

Now, note that

$$\|x\|_{L^2(\mu),\circ} = \int \mu(s) \left( \frac{\|z_s\|_\circ^*}{\mu(s)} \right)^2 \|z_s^*\|_\circ^2 = \int \frac{1}{\mu(s)} \left( \|z_s\|_\circ^* \right)^2 = \langle x, z \rangle,$$

hence, for $\bar{x} := x/\|x\|_{L^2(\mu),\circ}$ we have $\|\bar{x}\|_{L^2(\mu),\circ} = 1$, and

$$\langle \bar{x}, z \rangle = \sqrt{\int \frac{1}{\mu(s)} \left( \|z_s\|_\circ^* \right)^2 \mathrm{d}s}.$$

On the other hand, for any $v$ such that $\|v\|_{L^2(\mu),\circ} \le 1$, we have

$$\langle v, z \rangle = \int \langle v_s, z_s \rangle \, \mathrm{d}s = \int \left\langle \mu(s) v_s, \mu(s)^{-1} z_s \right\rangle \mathrm{d}s$$

$$\le \int \left\| \sqrt{\mu(s)} v_s \right\|_\circ \left\| \sqrt{\mu(s)^{-1}} z_s \right\|_\circ^* \mathrm{d}s$$

$$\le \sqrt{\int \mu(s) \|v_s\|_\circ^2 \, \mathrm{d}s} \sqrt{\int \mu(s)^{-1} \left( \|z_s\|_\circ^* \right)^2 \mathrm{d}s}$$

$$\le \sqrt{\int \mu(s)^{-1} \left( \|z_s\|_\circ^* \right)^2 \mathrm{d}s},$$

and the proof is complete. $\qquad\square$

**Lemma 12.** *Let $\mu \in \Delta(\mathcal{S})$, and consider the state-action norm $\|\cdot\|_{L^2(\mu),\circ}$. For any $W \in \mathbb{R}^{SA}$, we have*

$$\|\mu \circ W\|_{L^2(\mu),\circ}^* = \sqrt{\mathbb{E}_{s \sim \mu} \left( \|W_s\|_\circ^* \right)^2}$$

*Proof.* By Lemma 11,

$$\|\mu \circ W\|_{L^2(\mu),\circ}^* = \sqrt{\int \mu(s)^{-1} \left( \|\mu(s) W_s\|_\circ^* \right)^2} = \sqrt{\mathbb{E}_{s \sim \mu} \left( \|W_s\|_\circ^* \right)^2}. \qquad\square$$

**Lemma 13.** *Assume $h \colon \mathbb{R}^A \to \mathbb{R}$ is 1-strongly convex and has $L$-Lipschitz gradient w.r.t. $\|\cdot\|$. Let $\mu \in \Delta(\mathcal{S})$, and define $R_\mu(\pi) := \mathbb{E}_{s \sim \mu}[h(\pi_s)]$. Then*

1. $B_{R_\mu}(\pi, \tilde{\pi}) = \mathbb{E}_{s \sim \mu} B_R(\pi_s, \tilde{\pi}_s)$.

2. $R_\mu$ *is 1-strongly convex and has an $L$-Lipschitz gradient w.r.t. $\|\cdot\|_{L^2(\mu),\circ}$.*

*Proof.* We have

$$\forall s, \nabla R_\mu(\pi)_s = \mu(s) \nabla R(\pi_s) \in \mathbb{R}^A$$

$$\implies B_{R_\mu}(\pi, \tilde{\pi}) = R_\mu(\pi) - R_\mu(\tilde{\pi}) - \left\langle \nabla R_\mu(\tilde{\pi}), \pi - \tilde{\pi} \right\rangle$$

$$= \mathbb{E}_{s \sim \mu} \left[ R(\pi_s) - R(\tilde{\pi}_s) - \langle \nabla R(\tilde{\pi}_s), \pi_s - \tilde{\pi}_s \rangle \right]$$

$$= \mathbb{E}_{s \sim \mu} B_R(\pi_s, \tilde{\pi}_s).$$

Further, 1-strongly convexity follows by

$$\mathbb{E}_{s \sim \mu} B_R(\pi_s, \tilde{\pi}_s) \geq \frac{1}{2} \mathbb{E}_{s \sim \mu} \|\pi_s - \tilde{\pi}_s\|_\circ^2,$$

and the Lipschitz gradient condition from Lemma 12:

$$
\begin{aligned}
\left\| \nabla R_\mu(\pi) - \nabla R_\mu(\pi^+) \right\|_{L^2(\mu),\circ}^* &= \left\| \mu \circ \left( \nabla h(\pi_s) - \nabla h(\pi_s^+) \right) \right\|_{L^2(\mu),\circ}^* \\
&= \sqrt{\mathbb{E}_{s \sim \mu} \left( \|\nabla h(\pi_s) - \nabla h(\pi_s^+)\|_\circ^* \right)^2} \\
&\leq L \sqrt{\mathbb{E}_{s \sim \mu} \|\pi_s - \pi_s^+\|_\circ^2} \\
&= L \left\| \pi - \pi^+ \right\|_{L^2(\mu),\circ},
\end{aligned}
$$

which completes the proof. $\qquad\square$

## D. Deferred proofs

### D.1. Auxiliary Lemmas

**Lemma 14** (Value difference; Kakade & Langford, 2002)**.** *For any $\rho \in \Delta(\mathcal{S})$,*

$$V_\rho(\tilde{\pi}) - V_\rho(\pi) = \frac{1}{1 - \gamma} \mathbb{E}_{s \sim \mu_\rho^\pi} \left\langle Q_s^{\tilde{\pi}}, \tilde{\pi}_s - \pi_s \right\rangle.$$

**Lemma 15** (Policy gradient theorem; Sutton et al., 1999)**.** *For any $\rho \in \Delta(\mathcal{S})$,*

$$\left( \nabla V_\rho(\pi) \right)_{s,a} = \frac{1}{1 - \gamma} \mu_\rho^\pi(s) Q_{s,a}^\pi,$$

$$\left\langle \nabla V_\rho(\pi), \tilde{\pi} - \pi \right\rangle = \frac{1}{1 - \gamma} \mathbb{E}_{s \sim \mu_\rho^\pi} \left\langle Q_s^\pi, \tilde{\pi}_s - \pi_s \right\rangle.$$

The following lemma can be found in e.g., (Bhandari & Russo, 2024; Agarwal et al., 2021). The proof below is provided for convenience.

**Lemma 16.** *Let $\Pi \subset \Delta(\mathcal{A})^\mathcal{S}$, $\Pi_{\text{all}} := \Delta(\mathcal{A})^\mathcal{S}$ and suppose that for any policy $\pi \in \Pi$, we have*

$$\max_{\pi^+ \in \Pi} \mathbb{E}_{s \sim \mu^\pi} \left\langle Q^\pi, \pi - \pi^+ \right\rangle \geq \max_{\pi' \in \Pi_{\text{all}}} \mathbb{E}_{s \sim \mu^\pi} \left\langle Q^\pi, \pi - \pi' \right\rangle - \epsilon.$$

*Then $\Pi$ is $(H\nu_0, \epsilon H^2 \nu_0)$-VGD w.r.t. $\mathcal{M}$, for $\nu_0 := \left\| \frac{\mu^\star}{\rho_0} \right\|_\infty$.*

*Proof.* Let $\pi^\star \in \arg\min_{\pi \in \Pi} V(\pi)$. By value difference Lemma 14,

$$
\begin{aligned}
V(\pi) - V(\pi^\star) &= H \mathbb{E}_{s \sim \mu^\star} \left[ \left\langle Q_s^\pi, \pi_s - \pi_s^\star \right\rangle \right] \\
&\leq H \max_{\pi' \in \Pi_{\text{all}}} \mathbb{E}_{s \sim \mu^\star} \left[ \left\langle Q_s^\pi, \pi_s - \pi_s' \right\rangle \right] \\
&\overset{(*)}{\leq} H \left\| \frac{\mu^\star}{\mu^\pi} \right\|_\infty \max_{\pi' \in \Pi_{\text{all}}} \mathbb{E}_{s \sim \mu^\pi} \left[ \left\langle Q_s^\pi, \pi_s - \pi_s' \right\rangle \right] \\
&\leq H \left\| \frac{\mu^\star}{\mu^\pi} \right\|_\infty \max_{\pi^+ \in \Pi} \mathbb{E}_{s \sim \mu^\pi} \left[ \left\langle Q_s^\pi, \pi_s - \pi_s^+ \right\rangle \right] + \epsilon H \left\| \frac{\mu^\star}{\mu^\pi} \right\|_\infty \\
&= \left\| \frac{\mu^\star}{\mu^\pi} \right\|_\infty \max_{\pi^+ \in \Pi} \left\langle \nabla V^\pi, \pi - \pi^+ \right\rangle + \epsilon H \left\| \frac{\mu^\star}{\mu^\pi} \right\|_\infty \\
&\overset{(**)}{\leq} H \left\| \frac{\mu^\star}{\rho_0} \right\|_\infty \max_{z \in \Pi} \left\langle \nabla V^\pi, \pi - z \right\rangle + \epsilon H^2 \left\| \frac{\mu^\star}{\rho_0} \right\|_\infty.
\end{aligned}
$$

To explain the transitions above, $(*)$ follows by the fact that within the complete policy class we may choose $\pi'$ to be greedy w.r.t. $Q^\pi$, which means $\langle Q_s^\pi, \pi_s - \pi'_s \rangle \geq 0$ for all $s \in \mathcal{S}$. The last transition $(**)$ follows from the fact that:

$$\mu^\pi(s) = \frac{1}{H} \sum_{t=0}^\infty \Pr(s_t = s \mid \rho_0, \pi) = \frac{1}{H}\rho_0(s) + \sum_{t=1}^\infty \Pr(s_t = s \mid \rho_0, \pi) \geq \frac{1}{H}\rho_0(s). \qquad \square$$

**Lemma 17.** *For any policy* $\pi \colon \mathcal{S} \to \Delta(\mathcal{A})$, $s, a \in \mathcal{S} \times \mathcal{A}$:

$$Q_{s,a}^{\tilde{\pi}} - Q_{s,a}^{\pi} = \gamma H \mathbb{E}_{s' \sim \mu_{\mathbb{P}_{s,a}}^\pi} \left\langle Q_{s'}^{\tilde{\pi}}, \tilde{\pi}_{s'} - \pi_{s'} \right\rangle.$$

*Proof.* By Lemma 14, we have:

$$\begin{aligned}
Q_{s,a}^{\tilde{\pi}} - Q_{s,a}^{\pi} &= \gamma \mathbb{E}_{s' \sim \mathbb{P}_{s,a}} \left[ V^{\tilde{\pi}}(s') - V^\pi(s') \right] \\
&= \gamma H \mathbb{E}_{s' \sim \mathbb{P}_{s,a}} \left[ \mathbb{E}_{s'' \sim \mu_{s'}^\pi} \left\langle Q_{s''}^{\tilde{\pi}}, \tilde{\pi}_{s''} - \pi_{s''} \right\rangle \right] \\
&= \gamma H \sum_{s'} \mathbb{P}(s'|s, a) \sum_{s''} \mu_{s'}^\pi(s'') \left\langle Q_{s''}^{\tilde{\pi}}, \tilde{\pi}_{s''} - \pi_{s''} \right\rangle \\
&= \gamma H \sum_{s''} \sum_{s'} \mathbb{P}(s'|s, a) \mu_{s'}^\pi(s'') \left\langle Q_{s''}^{\tilde{\pi}}, \tilde{\pi}_{s''} - \pi_{s''} \right\rangle \\
&= \gamma H \sum_{s''} \mu_{\mathbb{P}_{s,a}}^\pi(s'') \left\langle Q_{s''}^{\tilde{\pi}}, \tilde{\pi}_{s''} - \pi_{s''} \right\rangle \\
&= \gamma H \mathbb{E}_{s'' \sim \mu_{\mathbb{P}_{s,a}}^\pi} \left\langle Q_{s''}^{\tilde{\pi}}, \tilde{\pi}_{s''} - \pi_{s''} \right\rangle. \qquad \square
\end{aligned}$$

**Lemma 18.** *Let* $h \colon \mathbb{R}^A \to \mathbb{R}$ *be the negative entropy regularizer* $h(p) \coloneqq \sum_i p_i \log p_i$, *and assume* $\Delta_\epsilon(\mathcal{A}) \subset \Delta(\mathcal{A})$ *is such that* $p_i \geq \epsilon$ *for all* $p \in \Delta_\epsilon(\mathcal{A})$. *Then* $h$ *has* $1/\epsilon$*-Lipschitz gradient w.r.t.* $\|\cdot\|_1$ *over* $\Delta_\epsilon(\mathcal{A})$.

*Proof.* Let $p, \tilde{p} \in \Delta_\epsilon(\mathcal{A})$, and note,

$$\|\nabla h(p) - \nabla h(\tilde{p})\|_1^* = \|\nabla h(p) - \nabla h(\tilde{p})\|_\infty.$$

Let $i \in \mathcal{A}$, and observe that by the mean value theorem, for some $\alpha \in [p_i, \tilde{p}_i]$,

$$|\log(p_i) - \log(\tilde{p}_i)| = \left| \frac{\partial \log(x)}{\partial x} \right|_{x=\alpha} |p_i - \tilde{p}_i| = \frac{1}{\alpha} |p_i - \tilde{p}_i| \leq \frac{1}{\epsilon} |p_i - \tilde{p}_i| \leq \frac{1}{\epsilon} \|p - \tilde{p}\|_1,$$

since $p_i, \tilde{p}_i \geq \epsilon$. $\qquad \square$

### D.2. Proof of Lemma 2

We have, by Lemmas 14 and 15,

$$\begin{aligned}
\left| V^{\tilde{\pi}} - V^\pi - \langle \nabla V^\pi, \tilde{\pi} - \pi \rangle \right| &= \left| H \mathbb{E}_{s \sim \mu^\pi} \left\langle Q_s^{\tilde{\pi}}, \tilde{\pi}_s - \pi_s \right\rangle - H \mathbb{E}_{s \sim \mu^\pi} \left\langle Q_s^\pi, \tilde{\pi}_s - \pi_s \right\rangle \right| \\
&= H \left| \mathbb{E}_{s \sim \mu^\pi} \left\langle Q_s^{\tilde{\pi}} - Q_s^\pi, \tilde{\pi}_s - \pi_s \right\rangle \right|.
\end{aligned}$$

Applying Lemma 17 yields,

$$
\frac{1}{\gamma H^2} \left| V^{\tilde{\pi}} - V^\pi - \langle \nabla V^\pi, \tilde{\pi} - \pi \rangle \right|
$$

$$
= \left| \mathbb{E}_{s \sim \mu^\pi} \left[ \sum_a \left( \mathbb{E}_{s' \sim \mu^\pi_{\mathbb{P}_{s,a}}} \left\langle Q^{\tilde{\pi}}_{s'}, \tilde{\pi}_{s'} - \pi_{s'} \right\rangle \right) (\tilde{\pi}_{sa} - \pi_{sa}) \right] \right|
$$

$$
= \left| \sum_s \mu^\pi(s) \sum_a (\tilde{\pi}_{sa} - \pi_{sa}) \left( \sum_{s'} \mu^\pi_{\mathbb{P}_{sa}}(s') \left\langle Q^{\tilde{\pi}}_{s'}, \tilde{\pi}_{s'} - \pi_{s'} \right\rangle \right) \right|
$$

$$
= \left| \sum_{s,a} \sqrt{\mu^\pi(s)} (\tilde{\pi}_{sa} - \pi_{sa}) \left( \sqrt{\mu^\pi(s)} \sum_{s'} \mu^\pi_{\mathbb{P}_{sa}}(s') \left\langle Q^{\tilde{\pi}}_{s'}, \tilde{\pi}_{s'} - \pi_{s'} \right\rangle \right) \right|
$$

$$
\leq \sqrt{\sum_{s,a} \mu^\pi(s) (\tilde{\pi}_{sa} - \pi_{sa})^2} \sqrt{\sum_{s,a} \mu^\pi(s) \left( \sum_{s'} \mu^\pi_{\mathbb{P}_{sa}}(s') \left\langle Q^{\tilde{\pi}}_{s'}, \tilde{\pi}_{s'} - \pi_{s'} \right\rangle \right)^2} \qquad \text{(Cauchy-Schwarz)}
$$

$$
\leq \sqrt{\sum_{s,a} \mu^\pi(s) (\tilde{\pi}_{sa} - \pi_{sa})^2} \sqrt{\sum_{s,a} \mu^\pi(s) \sum_{s'} \mu^\pi_{\mathbb{P}_{sa}}(s') \left\langle Q^{\tilde{\pi}}_{s'}, \tilde{\pi}_{s'} - \pi_{s'} \right\rangle^2} \qquad \text{(Jensen)}
$$

$$
= \sqrt{\sum_s \mu^\pi(s) \left\| \tilde{\pi}_s - \pi_s \right\|_2^2} \sqrt{\sum_{s'} \left( \sum_{s,a} \frac{1}{\pi_{sa}} \mu^\pi(s) \pi_{sa} \mu^\pi_{\mathbb{P}_{sa}}(s') \right) \left\langle Q^{\tilde{\pi}}_{s'}, \tilde{\pi}_{s'} - \pi_{s'} \right\rangle^2}
$$

$$
\leq \frac{1}{\sqrt{\epsilon}} \sqrt{\sum_s \mu^\pi(s) \left\| \tilde{\pi}_s - \pi_s \right\|_2^2} \sqrt{\sum_{s'} \left( \sum_{s,a} \mu^\pi(s) \pi_{sa} \mu^\pi_{\mathbb{P}_{sa}}(s') \right) \left\langle Q^{\tilde{\pi}}_{s'}, \tilde{\pi}_{s'} - \pi_{s'} \right\rangle^2},
$$

for $\epsilon := \min_{s,a} \{\pi_{sa}\}$. Now, by the law of total probability (applied on the discounted probability measure $\mu^\pi$):

$$
\sum_{s,a} \mu^\pi(s) \pi_{sa} \mu^\pi_{\mathbb{P}_{sa}}(s') = \sum_{s,a} \mu^\pi(s \mid s_0 \sim \rho_0) \pi(a|s) \mu^\pi(s' \mid s'_0 \sim \mathbb{P}_{sa})
$$

$$
= \sum_{s,a} \mu^\pi(s, a \mid s_0 \sim \rho_0) \mu^\pi(s' \mid s'_0 \sim \mathbb{P}_{sa})
$$

$$
= \mu^\pi(s' \mid s_0 \sim \rho_0)
$$

$$
= \mu^\pi(s').
$$

Combining with our previous inequality, we obtain

$$
\left| V^{\tilde{\pi}} - V^\pi - \langle \nabla V^\pi, \tilde{\pi} - \pi \rangle \right| \leq \frac{\gamma H^2}{\sqrt{\epsilon}} \sqrt{\sum_s \mu^\pi(s) \left\| \tilde{\pi}_s - \pi_s \right\|_2^2} \sqrt{\sum_{s'} \mu^\pi(s') \left\langle Q^{\tilde{\pi}}_{s'}, \tilde{\pi}_{s'} - \pi_{s'} \right\rangle^2}
$$

$$
= \frac{\gamma H^2}{\sqrt{\epsilon}} \left\| \tilde{\pi} - \pi \right\|_{L^2(\mu^\pi),2} \sqrt{\sum_{s'} \mu^\pi(s') \left\langle Q^{\tilde{\pi}}_{s'}, \tilde{\pi}_{s'} - \pi_{s'} \right\rangle^2}.
$$

Further,

$$
\sqrt{\sum_{s'} \mu^\pi(s') \left\langle Q^{\tilde{\pi}}_{s'}, \tilde{\pi}_{s'} - \pi_{s'} \right\rangle^2} \leq \sqrt{\sum_{s'} \mu^\pi(s') \left\| Q^{\tilde{\pi}}_{s'} \right\|_\infty^2 \left\| \tilde{\pi}_{s'} - \pi_{s'} \right\|_1^2} \leq H \left\| \tilde{\pi} - \pi \right\|_{L^2(\mu^\pi),1},
$$

and

$$
\sqrt{\sum_{s'} \mu^\pi(s') \left\langle Q^{\tilde{\pi}}_{s'}, \tilde{\pi}_{s'} - \pi_{s'} \right\rangle^2} \leq \sqrt{\sum_{s'} \mu^\pi(s') \left\| Q^{\tilde{\pi}}_{s'} \right\|_2^2 \left\| \tilde{\pi}_{s'} - \pi_{s'} \right\|_2^2} \leq A H \left\| \tilde{\pi} - \pi \right\|_{L^2(\mu^\pi),2}.
$$

The first inequality above gives

$$\left| V^{\tilde{\pi}} - V^{\pi} - \langle \nabla V^{\pi}, \tilde{\pi} - \pi \rangle \right| \leq \frac{\gamma H^3}{\sqrt{\epsilon}} \, \| \tilde{\pi} - \pi \|_{L^2(\mu^{\pi}),2} \, \| \tilde{\pi} - \pi \|_{L^2(\mu^{\pi}),1} \leq \frac{\gamma H^3}{\sqrt{\epsilon}} \, \| \tilde{\pi} - \pi \|^2_{L^2(\mu^{\pi}),1},$$

which proves the first claim, and the second one

$$\left| V^{\tilde{\pi}} - V^{\pi} - \langle \nabla V^{\pi}, \tilde{\pi} - \pi \rangle \right| \leq \frac{\gamma A H^3}{\sqrt{\epsilon}} \, \| \tilde{\pi} - \pi \|_{L^2(\mu^{\pi}),2} \, \| \tilde{\pi} - \pi \|_{L^2(\mu^{\pi}),2} = \frac{\gamma A H^3}{\sqrt{\epsilon}} \, \| \tilde{\pi} - \pi \|^2_{L^2(\mu^{\pi}),2},$$

which proves the second and completes the proof. □

### D.3. Proof of Theorem 1

The theorem makes use of the following.

**Lemma 19.** *Assume $\Pi$ is $(C_\star, \varepsilon_{\mathrm{vgd}})$-VGD w.r.t. $\mathcal{M}$, and consider the $\epsilon$-greedy exploratory version of $\Pi$, $\Pi^\epsilon :=$ $\{(1-\epsilon)\pi + \epsilon u \mid \pi \in \Pi\}$, where $u_{s,a} \equiv 1/A$. Then $\Pi^\epsilon$ is $(C_\star, \delta)$-VGD with $\delta := \varepsilon_{\mathrm{vgd}} + 12 C_\star H^2 A \epsilon$. Concretely, for any $\pi^\epsilon \in \Pi^\epsilon$, we have:*

$$C_\star \max_{\tilde{\pi}^\epsilon \in \Pi^\epsilon} \langle \nabla V(\pi^\epsilon), \tilde{\pi}^\epsilon - \pi^\epsilon \rangle \geq V(\pi^\epsilon) - V^\star(\Pi^\epsilon) - \varepsilon_{\mathrm{vgd}} - 12\epsilon C_\star H^2 A.$$

We now prove our corollary and return to prove the above lemma later in Appendix D.4.

*Proof of Theorem 1.* By Lemma 19, we have that $\Pi^{\varepsilon_{\mathrm{expl}}}$ is $(C_\star, \delta)$-VGD with $\delta = \varepsilon_{\mathrm{vgd}} + 12\varepsilon_{\mathrm{expl}} C_\star H^2 A$. Therefore, under the conditions of Theorem 3 and the value difference Lemma 14,

$$V(\pi^{K+1}) - V^\star(\Pi) \leq V(\pi^{K+1}) - V^\star(\Pi^{\varepsilon_{\mathrm{expl}}}) + \left| V^\star(\Pi^{\varepsilon_{\mathrm{expl}}}) - V^\star(\Pi) \right|$$

$$= O\left( \frac{C_\star^2 L^2 c_1^2}{\eta K} + \left( C_\star D + c_1 L^2 \right) H \sqrt{\varepsilon_{\mathrm{crit}}} + C_\star \varepsilon_{\mathrm{act}} + c_1 L \eta^{-1/2} \sqrt{\varepsilon_{\mathrm{act}}} + \delta \right),$$

where $c_1 := D + \eta HM$. Next we apply Lemma 4 in the both cases considered, using the fact that for all $\pi \in \Pi^{\varepsilon_{\mathrm{expl}}}$, we have $\min_{s,a} \{\pi_{s,a}\} \geq \varepsilon_{\mathrm{expl}}/A$. In the euclidean case, we argue the following:

1. $R$ is 1-strongly convex and has 1-Lipschitz gradient w.r.t. $\|\cdot\|_2$.

2. $\forall s, \|\pi_s - \tilde{\pi}_s\|_2 \leq D = 2, \|Q_s\|_2 \leq M = \sqrt{AH}$.

3. The value function is $\left( \beta := \frac{A^{3/2} H^3}{\sqrt{\varepsilon_{\mathrm{expl}}}} \right)$-locally smooth w.r.t. $\pi \mapsto \|\cdot\|_{L^2(\mu^\pi),2}$.

Hence, $c_1 = O(1)$, and Lemma 4 gives:

$$V(\pi^{K+1}) - V^\star(\Pi) \lesssim \frac{C_\star^2}{\eta K} + C_\star \left( H \sqrt{\varepsilon_{\mathrm{crit}}} + \varepsilon_{\mathrm{act}} \right) + \eta^{-1/2} \sqrt{\varepsilon_{\mathrm{act}}} + \delta$$

$$= \frac{2A^{3/2} H^3 C_\star^2}{\sqrt{\varepsilon_{\mathrm{expl}}} K} + C_\star \left( H \sqrt{\varepsilon_{\mathrm{crit}}} + \varepsilon_{\mathrm{act}} \right) + \frac{\sqrt{2A^{3/2} H^3}}{\varepsilon_{\mathrm{expl}}^{1/4}} \sqrt{\varepsilon_{\mathrm{act}}} + \delta.$$

Setting $\varepsilon_{\mathrm{expl}} = K^{-2/3}$, we obtain

$$V(\pi^{K+1}) - V^\star(\Pi) = O\left( \frac{C_\star^2 A^{3/2} H^3}{K^{2/3}} + C_\star \left( H \sqrt{\varepsilon_{\mathrm{crit}}} + \varepsilon_{\mathrm{act}} \right) + A H^2 K^{1/6} \sqrt{\varepsilon_{\mathrm{act}}} + \varepsilon_{\mathrm{vgd}} \right).$$

In the negative-entropy case, we have the following.

1. $R$ is 1-strongly convex and has a $(A/\varepsilon_{\mathrm{expl}})$-Lipschitz gradient w.r.t. $\|\cdot\|_1$ (by Pinsker's inequality and Lemma 18).

2. $\forall s, \|\pi_s - \tilde{\pi}_s\|_1 \le D = 2, \|Q_s\|_1 \le M = H.$

3. The value function is $\left(\beta := \frac{A^{1/2}H^3}{\sqrt{\varepsilon_{\text{expl}}}}\right)$-locally smooth w.r.t. $\pi \mapsto \|\cdot\|_{L^2(\mu^\pi),1}$.

Hence, $c_1 = O(1)$, and Lemma 4 gives:

$$
\begin{aligned}
V(\pi^{K+1}) - V^\star(\Pi) &\lesssim \frac{C_\star^2 A^2}{\varepsilon_{\text{expl}}^2 \eta K} + \left(C_\star + \frac{A^2}{\varepsilon_{\text{expl}}^2}\right)H\sqrt{\varepsilon_{\text{crit}}} + C_\star \varepsilon_{\text{act}} + \frac{A}{\varepsilon_{\text{expl}}\sqrt{\eta}}\sqrt{\varepsilon_{\text{act}}} + \delta \\
&= \frac{2A^{5/2}H^3 C_\star^2}{\varepsilon_{\text{expl}}^{5/2} K} + \left(C_\star + \frac{A^2}{\varepsilon_{\text{expl}}^2}\right)H\sqrt{\varepsilon_{\text{crit}}} + C_\star \varepsilon_{\text{act}} + \frac{A^{3/2}H^3}{\varepsilon_{\text{expl}}^{5/4}}\sqrt{\varepsilon_{\text{act}}} + \delta.
\end{aligned}
$$

We now set $\varepsilon_{\text{expl}} = K^{-2/7}A^{2/5}$ in order to balance the terms,

$$
\frac{2A^{5/2}H^3 C_\star^2}{\varepsilon_{\text{expl}}^{5/2} K} + C_\star H^2 A \varepsilon_{\text{expl}},
$$

which yields,

$$
\begin{aligned}
&V(\pi^{K+1}) - V^\star(\Pi) \\
&= O\left(\frac{C_\star^2 A^{3/2}H^3}{K^{2/7}} + \left(C_\star + A^2 K^{4/7}\right)H\sqrt{\varepsilon_{\text{crit}}} + C_\star \varepsilon_{\text{act}} + A^{3/2}H^3 K^{5/14}\sqrt{\varepsilon_{\text{act}}} + \varepsilon_{\text{vgd}}\right),
\end{aligned}
$$

and completes the proof. $\qquad\square$

### D.4. Proof of Lemma 19

**Lemma 20.** *For any MDP $\mathcal{M} = (\mathcal{S}, \mathcal{A}, \mathbb{P}, \ell, \gamma, \rho_0)$ and two policies $\pi, \tilde{\pi}: \mathcal{S} \to \Delta(\mathcal{A})$, we have:*

$$
\left\|\mu^{\tilde{\pi}} - \mu^\pi\right\|_1 \le H\|\tilde{\pi} - \pi\|_{L^1(\mu^\pi),1}.
$$

*Proof.* Consider the MDP $\mathcal{M}_x = (\mathcal{S}, \mathcal{A}, \mathbb{P}, r_x, \gamma, \rho_0)$; i.e., the same MDP $\mathcal{M}$ but with reward function defined by $r_x(s, a) := \mathbb{1}\{s = x\}$. Let $V_{\cdot;r_x}, Q_{\cdot,\cdot;r_x}$ denote its value and action-value functions, respectively. We have

$$
\begin{aligned}
Q_{s,a;r_x}^{\tilde{\pi}} &= \mathbb{E}\left[\sum_{t=0}^\infty \gamma^t \mathbb{1}\{s_t = x\} \mid s_0 = s, a_0 = a, \tilde{\pi}\right] \\
&= \sum_{t=0}^\infty \gamma^t \Pr(s_t = x \mid s_0 = s, a_0 = a, \tilde{\pi}) \\
&= \mathbb{1}\{s = x\} + \sum_{t=1}^\infty \gamma^t \Pr(s_t = x \mid s_0 = s, a_0 = a, \tilde{\pi}) \\
&= \mathbb{1}\{s = x\} + \gamma \sum_{t=1}^\infty \gamma^{t-1} \Pr(s_t = x \mid s_1 \sim \mathbb{P}_{sa}, \tilde{\pi}) \\
&= \mathbb{1}\{s = x\} + \gamma \mu_{\mathbb{P}_{sa}}^{\tilde{\pi}}(x).
\end{aligned}
$$

Hence,

$$
\begin{aligned}
\mu^{\tilde{\pi}}(x) - \mu^{\pi}(x) &= V^{\tilde{\pi}}_{\rho_0; r_x} - V^{\pi}_{\rho_0; r_x} \\
&= H \mathbb{E}_{s \sim \mu^{\pi}} \left\langle Q^{\tilde{\pi}}_{s; r_x}, \tilde{\pi}_s - \pi_s \right\rangle \qquad \text{(Lemma 14)} \\
&= H \mathbb{E}_{s \sim \mu^{\pi}} \left[ \sum_a \left( \mathbb{I}\{s = x\} + \gamma \mu^{\tilde{\pi}}_{\mathbb{P}_{sa}}(x) \right) (\tilde{\pi}_{sa} - \pi_{sa}) \right] \\
&= H \mathbb{E}_{s \sim \mu^{\pi}} \left[ \sum_a \mathbb{I}\{s = x\} (\tilde{\pi}_{sa} - \pi_{sa}) + \gamma \sum_a \mu^{\tilde{\pi}}_{\mathbb{P}_{sa}}(x) (\tilde{\pi}_{sa} - \pi_{sa}) \right] \\
&= \gamma H \mathbb{E}_{s \sim \mu^{\pi}} \left[ \sum_a \mu^{\tilde{\pi}}_{\mathbb{P}_{sa}}(x) (\tilde{\pi}_{sa} - \pi_{sa}) \right].
\end{aligned}
$$

Therefore,

$$
\begin{aligned}
\sum_x \left| \mu^{\tilde{\pi}}(x) - \mu^{\pi}(x) \right| &= \gamma H \sum_x \left| \mathbb{E}_{s \sim \mu^{\pi}} \left[ \sum_a \mu^{\tilde{\pi}}_{\mathbb{P}_{sa}}(x) (\tilde{\pi}_{sa} - \pi_{sa}) \right] \right| \\
&\leq \gamma H \sum_x \mathbb{E}_{s \sim \mu^{\pi}} \left[ \sum_a \mu^{\tilde{\pi}}_{\mathbb{P}_{sa}}(x) |\tilde{\pi}_{sa} - \pi_{sa}| \right] \\
&= \gamma H \mathbb{E}_{s \sim \mu^{\pi}} \left[ \sum_a \left( \sum_x \mu^{\tilde{\pi}}_{\mathbb{P}_{sa}}(x) \right) |\tilde{\pi}_{sa} - \pi_{sa}| \right] \\
&= \gamma H \mathbb{E}_{s \sim \mu^{\pi}} \left[ \sum_a |\tilde{\pi}_{sa} - \pi_{sa}| \right] \\
&= \gamma H \|\tilde{\pi} - \pi\|_{L^1(\mu^{\pi}),1},
\end{aligned}
$$

and the proof is complete. □

*Proof of Lemma 19.* Let $\pi^{\epsilon} \in \Pi^{\epsilon}$, and set $\pi \in \Pi$ to be the non-exploratory version of $\pi^{\epsilon}$. We have, by Lemma 14:

$$
V^{\pi} - V^{\pi^{\epsilon}} = \mathbb{E}_{s \sim \mu^{\pi}} \left\langle Q^{\pi^{\epsilon}}_s, \pi_s - \pi^{\epsilon}_s \right\rangle = \epsilon \mathbb{E}_{s \sim \mu^{\pi}} \left\langle Q^{\pi^{\epsilon}}_s, \pi_s - u \right\rangle \leq 2\epsilon H. \tag{13}
$$

In addition,

$$
\begin{aligned}
\left\| \nabla V(\pi^{\epsilon}) - \nabla V(\pi) \right\|_1 &= \sum_s \left\| \mu^{\pi^{\epsilon}}(s) Q^{\pi^{\epsilon}}_s - \mu^{\pi}(s) Q^{\pi}_s \right\|_1 \\
&\leq \sum_s \left\| Q^{\pi^{\epsilon}}_s \right\|_1 \left| \mu^{\pi^{\epsilon}}(s) - \mu^{\pi}(s) \right| + \sum_s \mu^{\pi}(s) \left\| Q^{\pi}_s - Q^{\pi^{\epsilon}}_s \right\|_1 \\
&\leq A H \left\| \mu^{\pi^{\epsilon}} - \mu^{\pi} \right\|_1 + \sum_s \mu^{\pi}(s) \left\| Q^{\pi}_s - Q^{\pi^{\epsilon}}_s \right\|_1.
\end{aligned}
$$

To bound the first term, apply Lemma 20:

$$
A H \left\| \mu^{\pi^{\epsilon}} - \mu^{\pi} \right\|_1 \leq A H^2 \left\| \pi^{\epsilon} - \pi \right\|_{L^1(\mu^{\pi}),1} \leq \epsilon A H^2 \left\| \pi - u \right\|_{L^1(\mu^{\pi}),1} \leq 2\epsilon A H^2.
$$

To bound the second term, we have for any $\tilde{\pi}$:

$$
\begin{aligned}
\sum_s \mu^{\pi}(s) \left\| Q^{\pi}_s - Q^{\tilde{\pi}}_s \right\|_1 &\leq H^2 \sum_s \mu^{\pi}(s) \sum_a \sum_{s'} \mu^{\pi}_{\mathbb{P}_{s,a}}(s') \|\tilde{\pi}_{s'} - \pi_s\|_1 \\
&= H^2 A \sum_{s'} \sum_{s,a} \mu^{\pi}(s) \frac{1}{A} \mu^{\pi}_{\mathbb{P}_{s,a}}(s') \|\tilde{\pi}_{s'} - \pi_s\|_1 \\
&= H^2 A \|\tilde{\pi} - \pi\|_{L^1(\nu),1},
\end{aligned}
$$

where $\nu \in \mathbb{R}^{\mathcal{S}}$ is defined by

$$\nu(s') = \sum_{s,a} \mu^{\pi}(s) \frac{1}{A} \mu^{\pi}_{\mathbb{P}_{s,a}}(s').$$

By the law of total probability, $\nu \in \Delta(\mathcal{S})$ is in fact a state probability measure. Hence, we obtain

$$\sum_s \mu^{\pi}(s) \left\| Q_s^{\pi} - Q_s^{\pi^{\epsilon}} \right\|_1 \le H^2 A \left\| \pi^{\epsilon} - \pi \right\|_{L^1(\nu),1} = \epsilon H^2 A \left\| \pi - u \right\|_{L^1(\nu),1} \le 2\epsilon H^2 A.$$

The bounds on both terms, combined with the previous display now yields

$$\left\| \nabla V(\pi^{\epsilon}) - \nabla V(\pi) \right\|_1 \le 4\epsilon A H^2. \tag{14}$$

We now turn to apply Equations (13) and (14) to establish the claimed VGD condition. Let $\pi^{\epsilon} \in \Pi^{\epsilon}$ be an arbitrary $\epsilon$-greedy policy and $\pi \in \Pi$ the non-exploratory version of $\pi^{\epsilon}$. The assumption that $\Pi$ is $(C_{\star}, \varepsilon_{\mathrm{vgd}})$-VGD implies

$$\max_{\tilde{\pi} \in \Pi} \langle \nabla V(\pi), \tilde{\pi} - \pi \rangle \ge \frac{1}{C_{\star}} \left( V(\pi) - V^{\star}(\Pi) - \varepsilon_{\mathrm{vgd}} \right).$$

Let $\tilde{\pi} \in \Pi$ be the policy maximizing the LHS, and $\tilde{\pi}^{\epsilon} = (1 - \epsilon)\tilde{\pi} + \epsilon u \in \Pi^{\epsilon}$ its corresponding greedy exploration policy. We have,

$$\begin{aligned}
\langle \nabla V(\pi^{\epsilon}), \tilde{\pi}^{\epsilon} - \pi^{\epsilon} \rangle &= (1 - \epsilon) \langle \nabla V(\pi^{\epsilon}), \tilde{\pi} - \pi \rangle \\
&= (1 - \epsilon) \langle \nabla V(\pi), \tilde{\pi} - \pi \rangle + (1 - \epsilon) \langle \nabla V(\pi^{\epsilon}) - \nabla V(\pi), \tilde{\pi} - \pi \rangle \\
&\ge \frac{1 - \epsilon}{C_{\star}} \left( V(\pi) - V^{\star}(\Pi) - \varepsilon_{\mathrm{vgd}} \right) + (1 - \epsilon) \langle \nabla V(\pi^{\epsilon}) - \nabla V(\pi), \tilde{\pi} - \pi \rangle \\
&\ge \frac{1}{C_{\star}} \left( V(\pi) - V^{\star}(\Pi) - \varepsilon_{\mathrm{vgd}} \right) - 2 \left\| \nabla V(\pi^{\epsilon}) - \nabla V(\pi) \right\|_1 \\
&\ge \frac{1}{C_{\star}} \left( V(\pi) - V^{\star}(\Pi) - \varepsilon_{\mathrm{vgd}} \right) - 8\epsilon H^2 A \qquad \text{(Equation (14))} \\
&\ge \frac{1}{C_{\star}} \left( V(\pi^{\epsilon}) - V^{\star}(\Pi) - \varepsilon_{\mathrm{vgd}} - |V(\pi^{\epsilon}) - V(\pi)| \right) - 8\epsilon H^2 A \\
&\ge \frac{1}{C_{\star}} \left( V(\pi^{\epsilon}) - V^{\star}(\Pi) - \varepsilon_{\mathrm{vgd}} - 2\epsilon H \right) - 8\epsilon H^2 A \qquad \text{(Equation (13))} \\
&\ge \frac{1}{C_{\star}} \left( V(\pi^{\epsilon}) - V^{\star}(\Pi^{\epsilon}) - \varepsilon_{\mathrm{vgd}} - 4\epsilon H \right) - 8\epsilon H^2 A. \qquad \text{(Equation (13))}
\end{aligned}$$

(Indeed, we pay for the difference $V^{\star}(\Pi^{\epsilon}) - V^{\star}(\Pi)$ here, only to pay it again in the other direction later, but it is cleaner this way and results in only an extra constant numerical factor.) Therefore,

$$C_{\star} \max_{\hat{\pi}^{\epsilon} \in \Pi^{\epsilon}} \langle \nabla V(\pi^{\epsilon}), \hat{\pi}^{\epsilon} - \pi^{\epsilon} \rangle \ge V(\pi^{\epsilon}) - V^{\star}(\Pi^{\epsilon}) - \varepsilon_{\mathrm{vgd}} - 12\epsilon C_{\star} H^2 A,$$

which completes the proof. $\qquad \square$

## E. Constrained non-convex optimization for locally smooth objectives: Analysis

In this section, we provide the full technical details for Section 3.2. Recall that we consider the constrained optimization problem:

$$\min_{x \in \mathcal{X}} f(x), \tag{15}$$

where the decision set $\mathcal{X} \subseteq \mathbb{R}^d$ is convex and endowed with a local norm $x \mapsto \|\cdot\|_x$ (see Definition 5), and access to the objective is granted through an approximate first order oracle, as defined in Assumption 3. We assume $f : \mathcal{X} \to \mathbb{R}$ is differentiable and defined over an open domain $\mathbf{dom}\, f \subseteq \mathbb{R}^d$ that contains $\mathcal{X}$. We consider an approximate version of the

algorithm described in Equation (8), hence for the sake of rigor, we introduce some additional notation. Given any convex regularizer $h \colon \mathbb{R}^d \to \mathbb{R}$, we define a Bregman proximal point update with step-size $\eta > 0$ by:

$$T_\eta(x; h) := \underset{y \in X}{\arg\min} \left\{ \left\langle \widehat{\nabla} f(x), y \right\rangle + \frac{1}{\eta} B_h(y, x) \right\}, \tag{16}$$

and the set of $\varepsilon_{\text{opt}}$-approximate solutions by:

$$T_\eta^{\varepsilon_{\text{opt}}}(x; h) := \left\{ x^+ \in X \mid \forall z \in X : \left\langle \widehat{\nabla} f(x) + \frac{1}{\eta} \nabla B_h(x^+, x), z - x^+ \right\rangle \geq -\varepsilon_{\text{opt}} \right\}. \tag{17}$$

Now, the approximate version of our algorithm is given by:

$$k = 1, \ldots, K : \quad x_{k+1} \in T_\eta^{\varepsilon_{\text{opt}}}(x_k; R_{x_k}). \tag{18}$$

We recall our main theorem below.

***Theorem*** (restatement of Theorem 3). *Suppose that $f$ is $(C_\star, \varepsilon_{\text{vgd}})$-VGD as per Definition 4, and that $f^\star := \min_{x \in X} f(x) > -\infty$. Assume further that:*

(i) *The local regularizer $R_x$ is 1-strongly convex and has an $L$-Lipschitz gradient w.r.t. $\|\cdot\|_x$ for all $x \in X$.*

(ii) *For all $x \in X$, $\max_{u,v \in X} \|u - v\|_x \leq D$, and $\|\nabla f(x)\|_x^* \leq M$.*

(iii) *$f$ is $\beta$-locally smooth w.r.t. $x \mapsto \|\cdot\|_x$.*

*Then, for the algorithm described in Equation (18) we have following guarantee when $\eta \leq 1/(2\beta)$:*

$$f(x_{K+1}) - f^\star = O\left( \frac{C_\star^2 L^2 c_1^2}{\eta K} + \left( C_\star D + c_1 L^2 \right) \varepsilon_{\bar{\nu}} + C_\star \varepsilon_{\text{opt}} + c_1 L \eta^{-\frac{1}{2}} \sqrt{\varepsilon_{\text{opt}}} + \varepsilon_{\text{vgd}} \right)$$

*where $c_1 := D + \eta M$.*

Evidently, since the objective is not convex, standard mirror descent analyses are inadequate, and our analysis takes the proximal point update view of Equation (18). While there are numerous prior works that investigate non-euclidean proximal point methods for both convex and non-convex objective functions (e.g., Tseng, 2010; Ghadimi et al., 2016; Bauschke et al., 2017; Lu et al., 2018; Zhang & He, 2018; Fatkhullin & He, 2024; see also Beck, 2017) , non of them fit into the specific setting we study here. The notable differences being the use of *local* smoothness (Definition 6), and the goal of seeking convergence in function values for a non-convex objective by exploiting variational gradient dominance.

Our approach may be best described as one that adapts the work of Xiao (2022) to the non-euclidean (and, "local") setup, but without relying on the objective having a Lipschitz gradient (note that we do not claim our definition of local smoothness implies a Lipschitz gradient condition). Since Xiao (2022) relies on global smoothness of the objective w.r.t. the euclidean norm (as was established by Agarwal et al., 2021), their bounds inevitably scale with the size of the state-space $S$, which we want to avoid. Given any convex regularizer $h \colon \mathbb{R}^d \to \mathbb{R}$, we define Bregman gradient mapping by:

$$G_\eta(x, x^+; h) := \frac{1}{\eta} \left( \nabla h(x) - \nabla h(x^+) \right), \tag{19}$$

where $x^+ \in \mathbb{R}^d$ should be interpreted as an approximate proximal point update step, i.e., $x^+ \in T_\eta^{\varepsilon_{\text{opt}}}(x; h)$.

### E.1. Bregman prox: Descent and Stationarity

In this section we provide basic results relating to proximal point descent and stationarity conditions. Our first lemma is (roughly) a non-euclidean version of a similar lemma given in (Nesterov, 2013) for the euclidean case.

**Lemma 21** (Bregman proximal step descent). *Let $\|\cdot\|$ be a norm, and suppose $x \in X$ is such that*

$$\forall y \in X : |f(y) - f(x) - \langle \nabla f(x), y - x \rangle| \leq \frac{\beta}{2} \|y - x\|^2.$$

*Assume further that:*

1. $0 < \eta \le 1/(2\beta)$,

2. $h \colon \mathbb{R}^d \to \mathbb{R}$ is 1-strongly convex and has an $L_h$-Lipschitz gradient, w.r.t. $\|\cdot\|$.

3. $\left\| \widehat{\nabla} f(x) - \nabla f(x) \right\|_* \le \varepsilon_\nabla$.

*Then, for $x^+ \in T_\eta^{\varepsilon_{\mathrm{opt}}}(x; h)$ we have that:*

$$f(x^+) \le f(x) - \frac{\eta}{2L_h^2} \left\| G_\eta(x, x^+; h) \right\|_*^2 + \eta \varepsilon_\nabla \left\| G_\eta(x, x^+; h) \right\|_* + \varepsilon_{\mathrm{opt}}.$$

*Proof.* Observe,

$$
\begin{aligned}
f(x^+) &\le f(x) + \left\langle \nabla f(x), x^+ - x \right\rangle + \frac{\beta}{2} \left\| x^+ - x \right\|^2 \\
&= f(x) + \left\langle \widehat{\nabla} f(x), x^+ - x \right\rangle + \frac{\beta}{2} \left\| x^+ - x \right\|^2 + \left\langle \nabla f(x) - \widehat{\nabla} f(x), x^+ - x \right\rangle \\
&\le f(x) + \left\langle \widehat{\nabla} f(x), x^+ - x \right\rangle + \frac{\beta}{2} \left\| x^+ - x \right\|^2 + \varepsilon_\nabla \left\| x^+ - x \right\| \\
&\le f(x) + \left\langle \widehat{\nabla} f(x), x^+ - x \right\rangle + \frac{\beta}{2} \left\| x^+ - x \right\|^2 + \eta \varepsilon_\nabla \left\| G_\eta(x, x^+; h) \right\|_* . \qquad \text{(Lemma 23)}
\end{aligned}
$$

Further, since $x^+ \in T_\eta^{\varepsilon_{\mathrm{opt}}}(x; h)$, for any $z \in \mathcal{X}$,

$$\left\langle \widehat{\nabla} f(x), x^+ - z \right\rangle \le \left\langle \frac{1}{\eta} \left( \nabla h(x^+) - \nabla h(x) \right), z - x^+ \right\rangle + \varepsilon_{\mathrm{opt}}.$$

Hence,

$$
\begin{aligned}
& f(x) + \left\langle \widehat{\nabla} f(x), x^+ - x \right\rangle + \frac{\beta}{2} \left\| x^+ - x \right\|^2 \\
&\le f(x) + \frac{1}{\eta} \left\langle \nabla h(x) - \nabla h(x^+), x^+ - x \right\rangle + \frac{\beta}{2} \left\| x^+ - x \right\|^2 + \varepsilon_{\mathrm{opt}} \\
&= f(x) - \frac{1}{\eta} \left( B_h(x^+, x) + B_h(x, x^+) \right) + \frac{\beta}{2} \left\| x^+ - x \right\|^2 + \varepsilon_{\mathrm{opt}} \\
&\le f(x) - \frac{1}{\eta} \left( B_h(x^+, x) + B_h(x, x^+) \right) + \beta B_h(x^+, x) + \varepsilon_{\mathrm{opt}} \\
&\le f(x) - \frac{1}{2\eta} \left( B_h(x^+, x) + B_h(x, x^+) \right) + \varepsilon_{\mathrm{opt}},
\end{aligned}
$$

where the last line inequality follows from $\eta \le 1/(2\beta)$. Combining with the previous derivation, we now have

$$f(x^+) \le f(x) - \frac{1}{2\eta} \left( B_h(x^+, x) + B_h(x, x^+) \right) + \varepsilon_{\mathrm{opt}} + \eta \varepsilon_\nabla \left\| G_\eta(x, x^+; h) \right\|_* . \qquad (20)$$

Finally, the assumption that $h$ has an $L_h$-Lipschitz gradient implies that

$$\eta^2 \left\| G_\eta(x, x^+; h) \right\|_*^2 = \left\| \nabla h(x^+) - \nabla h(x) \right\|_*^2 \le L_h^2 \left\| x^+ - x \right\|^2 \le 2L_h^2 B_h(x^+, x),$$

and similarly $\eta^2 \left\| G_\eta(x, x^+; h) \right\|_*^2 \le 2L_h^2 B_h(x, x^+)$. Hence,

$$-\frac{1}{2\eta} \left( B_h(x^+, x) + B_h(x, x^+) \right) \le -\frac{\eta}{2L_h^2} \left\| G_\eta(x, x^+; h) \right\|_*^2,$$

which completes the proof after combining with Equation (20). $\qquad \square$

Our second lemma bounds the error in optimality conditions at any point $x \in \mathcal{X}$ w.r.t. the gradient mapping dual norm. We remark that here we do not assume a Lipschitz gradient condition holds for the objective function, as commonly done in similar arguments (e.g., Nesterov, 2013; Xiao, 2022).

**Lemma 22.** *Let $\|\cdot\|$ be a norm, and $x \in \mathcal{X}$. Assume that:*

1. *$h \colon \mathbb{R}^d \to \mathbb{R}$ is 1-strongly convex and has an $L_h$-Lipschitz gradient, w.r.t. $\|\cdot\|$.*

2. *$\left\| \widehat{\nabla} f(x) - \nabla f(x) \right\|_* \leq \varepsilon_\nabla$,*

3. *$D > 0$ upper bounds the diameter of $\mathcal{X}$: $\max_{z,y \in \mathcal{X}} \|z - y\| \leq D$*

4. *$M > 0$ upper bounds the gradient dual norm at $x$: $\|\nabla f(x)\|_* \leq M$.*

*Then, if $x^+ \in T_\eta^{\varepsilon_{\mathrm{opt}}}(x; h)$, it holds that:*

$$\forall y \in \mathcal{X} : \langle \nabla f(x), x - y \rangle \leq (D + \eta M) \left\| G_\eta(x, x^+; h) \right\|_* + \varepsilon_\nabla D + \varepsilon_{\mathrm{opt}}.$$

*Proof.* By assumption, we have for all $y \in \mathcal{X}$,

$$\left\langle \widehat{\nabla} f(x) - G_\eta(x, x^+; h), y - x^+ \right\rangle \geq -\varepsilon_{\mathrm{opt}}$$

$$\iff \langle \nabla f(x), x^+ - y \rangle \leq \left\langle G_\eta(x, x^+; h), y - x^+ \right\rangle + \left\langle \nabla f(x) - \widehat{\nabla} f(x), x^+ - y \right\rangle + \varepsilon_{\mathrm{opt}}$$

$$\leq \left\| G_\eta(x, x^+; h) \right\|_* D + \varepsilon_\nabla D + \varepsilon_{\mathrm{opt}}.$$

Further,

$$
\begin{aligned}
\langle \nabla f(x), x - y \rangle &= \left\langle \nabla f(x), x^+ - y \right\rangle + \left\langle \nabla f(x), x - x^+ \right\rangle \\
&\leq \left\| G_\eta(x, x^+; h) \right\|_* D + \varepsilon_\nabla D + \varepsilon_{\mathrm{opt}} + \left\langle \nabla f(x), x - x^+ \right\rangle \\
&\leq \left\| G_\eta(x, x^+; h) \right\|_* D + \varepsilon_\nabla D + \varepsilon_{\mathrm{opt}} + M \left\| x - x^+ \right\| \\
&\leq \left\| G_\eta(x, x^+; h) \right\|_* D + \varepsilon_\nabla D + \varepsilon_{\mathrm{opt}} + \eta M \left\| G_\eta(x, x^+; h) \right\|_* \qquad \text{(Lemma 23)} \\
&\leq (D + \eta M) \left\| G_\eta(x, x^+; h) \right\|_* + \varepsilon_\nabla D + \varepsilon_{\mathrm{opt}},
\end{aligned}
$$

which completes the proof. $\qquad \square$

**Lemma 23.** *For any norm $\|\cdot\|$, and any $x, x^+ \in \mathcal{X}$, we have $\|x - x^+\| \leq \eta \left\| G_\eta(x, x^+; h) \right\|_*$.*

*Proof.* For any $u, v$ it holds that (see e.g., Hiriart-Urruty & Lemaréchal, 2004),

$$\frac{1}{2} \|u - v\|^2 \leq B_h(u, v) = B_{h^*}(\nabla h(u), \nabla h(v)) \leq \frac{1}{2} \|\nabla h(u) - \nabla h(v)\|_*^2 .$$

The result now follows by the definition of $G_\eta(x, x^+; h)$. $\qquad \square$

### E.2. Proof of Theorem 3

We begin by establishing the objective satisfies a weak gradient mapping domination condition similar (but not identical, due to the differences mentioned above) to that considered in Xiao (2022).

**Definition 8.** We say that $f \colon \mathcal{X} \to \mathbb{R}$ satisfies a weak gradient mapping domination condition w.r.t. a local regularizer $R$ if there exist $\delta, \omega > 0$ such that for all $x \in \mathcal{X}$:

$$\left\| G_\eta(x, x^+; h) \right\|_x^* \geq \sqrt{2\omega}(f(x) - f^\star - \delta)$$

The lemma below establishes our objective function satisfies Definition 8 with a suitable choice of parameters.

**Lemma 24.** *Suppose that $f$ is $(C_\star, \varepsilon_{\mathrm{vgd}})$-VGD as per Definition 4, and that $f^\star := \min_{x \in X} f(x) > -\infty$. Assume further that $R_x$ is 1-strongly convex and has an $L$-Lipschitz gradient w.r.t. $\|\cdot\|_x$ for all $x \in X$. Then, we have the following weak gradient mapping domination condition; for all $x \in X, x^+ \in T_\eta^{\varepsilon_{\mathrm{opt}}}(x; R_x)$:*

$$\left\| G_\eta(x, x^+; R_x) \right\|_x^* \geq \sqrt{2\omega} \left( f(x) - f^\star - \delta \right),$$

*for $\omega := \frac{1}{2} \left( C_\star(D + \eta M) \right)^{-2}$, $\delta := \varepsilon_{\mathrm{vgd}} + \varepsilon_{\mathrm{opt}} C_\star + \varepsilon_\nabla C_\star D$.*

*Proof.* Let $x \in X$, and apply Lemma 22 with $\|\cdot\| = \|\cdot\|_x$ and $h = R_x$, to obtain:

$$\forall y \in X : \langle \nabla f(x), x - y \rangle \leq (D + \eta M) \left\| G_\eta(x, x^+; R_x) \right\|_x^* + \varepsilon_\nabla D + \varepsilon_{\mathrm{opt}}.$$

Further, since $f$ is $(C_\star, \varepsilon_{\mathrm{vgd}})$-VGD, we have

$$\max_{y \in X} \langle \nabla f(x), x - y \rangle \geq \frac{1}{C_\star} \left( f(x) - f^\star - \varepsilon_{\mathrm{vgd}} \right).$$

Combining both inequalities, the result follows. $\square$

We are now ready to prove Theorem 3.

*Proof of Theorem 3.* In the sake of notational clarity, define:

$$\mathcal{G}_k := \left\| G_\eta(x_k, x_{k+1}; R_{x_k}) \right\|_{x_k}^*. \tag{21}$$

We begin by applying Lemma 21 for every $k \in [K]$ with $\|\cdot\| = \|\cdot\|_{x_k}$ and $h = R_{x_k}$, which implies,

$$f(x_{k+1}) - f(x_k) \leq -\frac{\eta}{2L^2} \mathcal{G}_k^2 + \eta \varepsilon_\nabla \mathcal{G}_k + \varepsilon_{\mathrm{opt}}. \tag{22}$$

Let us first assume that for all $k \in [K]$:

$$8L^2 \varepsilon_\nabla + \frac{4L}{\sqrt{\eta}} \sqrt{\varepsilon_{\mathrm{opt}}} \leq \mathcal{G}_k. \tag{23}$$

Then Equation (22) along with Lemma 24 gives

$$f(x_{k+1}) - f(x_k) \leq -\frac{\eta}{4L_h^2} \mathcal{G}_k^2 \leq -\frac{\eta\omega}{4L^2} \left( f(x_k) - f^\star - \delta \right)^2,$$

with $\omega := \frac{1}{2} \left( C_\star(D + \eta M) \right)^{-2}$ and $\delta := \varepsilon_{\mathrm{vgd}} + \varepsilon_{\mathrm{opt}} C_\star + \varepsilon_\nabla C_\star D$. We proceed to define $E_k := f(x_k) - f^\star$, and note that the above display implies $E_{k+1} \leq E_k$. Hence, we may assume that $E_k \geq 2\delta$ for all $k \in [K]$, otherwise the claim holds trivially. With this in mind, we now have,

$$E_{k+1} - E_k \leq -\frac{\eta\omega}{4L^2} (E_k - \delta)^2 \leq -\frac{\eta\omega}{16L^2} E_k^2.$$

Dividing both sides by $E_k E_{k+1}$ yields

$$\frac{1}{E_k} - \frac{1}{E_{k+1}} \leq -\frac{\eta\omega}{16L^2} \frac{E_k}{E_{k+1}}.$$

Summing over $k = 1, \ldots, K$ and telescoping the sum on the LHS, we obtain

$$\frac{1}{E_1} - \frac{1}{E_{K+1}} \leq -\frac{\eta\omega}{16L^2} \sum_{k=1}^K \frac{E_k}{E_{k+1}}$$

$$\iff E_{K+1} - E_1 \leq -\frac{\eta\omega}{16L^2} (E_{K+1} E_1) \sum_{k=1}^K \frac{E_k}{E_{k+1}} \leq -\frac{\eta\omega}{16L^2} (E_{K+1} E_1) K,$$

where the last inequality follows from the descent property $E_{k+1} \le E_k$. Rearranging, we now have

$$0 \le E_{K+1} \le E_1 \left(1 - \frac{\eta \omega}{16L^2} E_{K+1} K\right)$$

$$\implies E_{K+1} \le \frac{16L^2}{\eta \omega K} = \frac{32 C_\star^2 L^2 (D + \eta M)^2}{\eta K},$$

which completes the proof for the case that Equation (23) holds for all $k \in [K]$. Assume now that this is not the case, and let $k_0 \in [K]$ be the last iteration such that

$$\mathcal{G}_{k_0} < 8L^2 \varepsilon_\nabla + \frac{4L}{\sqrt{\eta}} \sqrt{\varepsilon_{\text{opt}}}.$$

Then by Lemma 22,

$$E_{k_0} \le (D + \eta M) \mathcal{G}_{k_0} + \varepsilon_\nabla D + \varepsilon_{\text{opt}} \le 8(D + \eta M) \left(L^2 \varepsilon_\nabla + L \sqrt{\varepsilon_{\text{opt}}/\eta}\right) + \varepsilon_\nabla D + \varepsilon_{\text{opt}},$$

and therefore by Equation (22),

$$E_{k_0+1} \le E_{k_0} + \eta \varepsilon_\nabla \mathcal{G}_{k_0} = O\left((D + \eta M) \left(L^2 \varepsilon_\nabla + L \sqrt{\varepsilon_{\text{opt}}/\eta}\right)\right).$$

Now, if $k_0 = K$ we are done. Otherwise, by the definition of $k_0$ we have that Equation (23) holds for all $k \in [k_0 + 1, K]$, hence $E_{k+1} \le E_k$ for all $k \ge k_0 + 1$. This implies that $E_{K+1} \le E_{k_0+1}$, which completes the proof. $\qquad\square$

### E.3. Convergence to stationary point without a VGD condition

In this section, we include a proof that the proximal point algorithm we consider converges to a stationary point, also without assuming a VGD condition. The proof follows from standard arguments and is given for completeness; for simplicity, we provide analysis only for the error free case. As an implication, we have that PMD converges to a stationary point in any MDP; this follows by combining the below theorem with Lemma 4 and Lemma 2, and proceeding with an argument similar to that of Theorem 1.

**Theorem 25.** *Suppose that $f^\star := \min_{x \in \mathcal{X}} f(x) > -\infty$, and assume:*

(i) *The local regularizer $R_x$ is 1-strongly convex and has an $L$-Lipschitz gradient w.r.t. $\|\cdot\|_x$ for all $x \in \mathcal{X}$.*

(ii) *For all $x \in \mathcal{X}$, $\max_{u,v \in \mathcal{X}} \|u - v\|_x \le D$, and $\|\nabla f(x)\|_x^* \le M$.*

(iii) *$f$ is $\beta$-locally smooth w.r.t. $x \mapsto \|\cdot\|_x$.*

*Consider an exact version of the proximal point algorithm Equation (8) with $\eta = 1/(2\beta)$ where $\varepsilon_\nabla = 0$ and $x^{k+1} = T_\eta(x_k; R_{x_k})$ for all $k$. Then, after $K$ iterations, there exists $k^\star \in [K]$ such that:*

$$\forall y \in \mathcal{X}, \quad \langle \nabla f(x_{k^\star}), y - x_{k^\star} \rangle \ge -\frac{2(D + \eta M) L \sqrt{\beta (f(x_1) - f(x^\star))}}{\sqrt{K}},$$

*Proof.* In the sake of notational clarity, define:

$$\mathcal{G}_k := \left\| G_\eta(x_k, x_{k+1}; R_{x_k}) \right\|_{x_k}^*.$$

We begin by applying Lemma 21 for every $k \in [K]$ with $\|\cdot\| = \|\cdot\|_{x_k}$ and $h = R_{x_k}$, which implies,

$$f(x_{k+1}) - f(x_k) \le -\frac{\eta}{2L^2} \mathcal{G}_k^2. \tag{24}$$

Now,

$$f(x_{K+1}) - f(x_1) = \sum_{k=1}^{K} f(x_{k+1}) - f(x_k) \leq -\frac{\eta}{2L^2} \sum_{k=1}^{K} \mathcal{G}_k^2,$$

thus, rearranging and bounding $f(x_{K+1}) \geq f(x^\star)$ gives

$$\frac{1}{K} \sum_{t=1}^{T} \mathcal{G}_k^2 \leq \frac{2L^2 \left( f(x_1) - f(x^\star) \right)}{\eta K}.$$

Hence, it must hold for $k^\star := \arg\min_k \mathcal{G}_k^2$;

$$\mathcal{G}_{k^\star}^2 \leq \frac{2L^2 \left( f(x_1) - f(x^\star) \right)}{\eta K}.$$

We now apply Lemma 22 to conclude,

$$\forall y \in X, \quad \langle \nabla f(x_{k^\star}), x_{k^\star} - y \rangle \leq \frac{(D + \eta M)L\sqrt{2 \left( f(x_1) - f(x^\star) \right)}}{\sqrt{\eta K}},$$

which implies the required result.

$\square$

## F. Policy Classes with Dual Parametrizations

In general, solving the following OMD problem in some state $s \in \mathcal{S}$,

$$\pi_s^{k+1} \leftarrow \underset{p \in \Delta(\mathcal{A})}{\arg\min} \left\langle Q_s^k, p \right\rangle + \frac{1}{\eta} B_R(p, \pi_s^k) \tag{25}$$

is equivalent to the following two updates:

$$\nabla R(\tilde{\pi}_s^{k+1}) \leftarrow \nabla R(\pi_s^k) - \eta Q_s^k$$
$$\pi_s^{k+1} = \Pi_{\Delta(\mathcal{A})}^R \left( \tilde{\pi}_s^{k+1} \right).$$

Let us denote the composition of the dual-to-primal mirror-map and the projection:

$$P_R(y) := \Pi_{\Delta(\mathcal{A})}^R \left( \nabla R^*(y) \right),$$

and note that

$$\pi_s^{k+1} = P_R(\nabla R(\tilde{\pi}_s^{k+1})).$$

When we are in a non-tabular setup and have a non-complete policy class $\Pi \neq \Delta(\mathcal{A})^{\mathcal{S}}$, we cannot update each state independently according to Equation (25). There are however a number of places we can "intervene" in the policy class representation to derive slightly different update procedures based on the dual variables. The PMD step in its general form is given by:

$$\pi^{k+1} \leftarrow \underset{\pi \in \Pi}{\arg\min} \, \mathbb{E}_{s \sim \mu^k} \left[ \left\langle Q_s^k, \pi_s \right\rangle + \frac{1}{\eta} B_R(\pi_s, \pi_s^k) \right] \tag{26}$$

Without making any assumptions regarding the parametric form of $\Pi$, we cannot decompose Equation (26) into meaningful dual space steps. We discuss next two types of policy class parameterizations and the update steps associated with them.

### F.1. Generic dual parameterizations

This is the approach taken in (Alfano et al., 2023) (see also the followup Xiong et al., 2024), and perhaps the most general one that allows for an explicit dual space update as well as leads to an approximate solution of Equation (26) that satisfies approximate optimality conditions in the complete-class setting. Consider a parametric function class $\mathcal{F}_\Theta := \{f_\theta \in \mathbb{R}^{SA} \mid \theta \in \Theta\}$, and the policy class:

$$\Pi(\mathcal{F}) := \{\pi^f \mid f \in \mathcal{F}_\Theta\}, \quad \text{where } \pi_s^f := P_R(f_s), \ \forall s \in \mathcal{S}.$$

Then, to solve Equation (26) we can proceed by:

$$f^{k+1} \leftarrow \underset{f \in \mathcal{F}}{\arg\min} \, \mathbb{E}_{s \sim \mu^k} \left[ \left\| f_s - \nabla R(\pi_s^k) - \eta Q_s^k \right\|_2^2 \right]$$

$$\pi^{k+1} \leftarrow \text{ the policy defined by } \pi_s^{k+1} = P_R(f_s^{k+1}) \tag{A}$$

### F.2. The log-linear policy class

This is a special case of the one discussed in the previous sub-section. In general, when we try to approximate the true solution of the unconstrained mirror descent step in a specific state:

$$f_s \approx \nabla R(\pi_s^k) - \eta Q_s^k,$$

we need to overcome two sources of error; one from the previous policy dual variable and one from the $Q$ function. More specifically, in general we have $\nabla R(\pi^k) \notin \mathcal{F}$ and $Q^k \notin \mathcal{F}$. (For $\pi \in \mathbb{R}^{SA}$ we define $\nabla R(\pi)_s := \nabla R(\pi_s)$.) In the special case that our function class $\mathcal{F}$ can represent $\nabla R(\pi)$ perfectly for all $\pi \in \Pi(\mathcal{F})$ and is closed to linear combinations, we can focus our attention on approximating the $Q$ function. Now, we may proceed by the following special case of $(A)$:

$$\widehat{Q}^k \leftarrow \underset{\widehat{Q} \in \mathcal{F}}{\arg\min} \, \mathbb{E}_{s,a \sim \mu^k} \left[ \left( \widehat{Q}_{s,a} - Q_{s,a}^k \right)^2 \right]$$

$$f^{k+1} \leftarrow \nabla R(\pi^k) - \eta \widehat{Q}^k$$

$$\pi^{k+1} \leftarrow \text{ the policy defined by } \pi_s^{k+1} = P_R(f_s^{k+1}). \tag{B}$$

Let $\phi_{s,a} \in \mathbb{R}^p$ be given feature vectors, and let $\phi_s := [\phi_{s,a_1} \cdots \phi_{s,a_A}] \in \mathbb{R}^{p \times A}$, and consider the log-linear policy class:

$$\Pi := \{\pi^\theta \mid \theta \in \mathbb{R}^p\}$$

$$\text{where } \forall s \in \mathcal{S}, \ \pi_s^\theta := P_R(\phi_s^\top \theta) = \frac{e^{\phi_s^\top \theta}}{\sum_a e^{\phi_{s,a}^\top \theta}}.$$

Note that:

1. This is the class $\Pi(\mathcal{F})$ for $\mathcal{F} = \{\theta \mapsto ((s,a) \mapsto \phi_{s,a}^\top \theta)\}$.

2. This is precisely a case where $\mathcal{F}$ can model $\nabla R(\pi)$ if $R$ is the negative entropy regularizer.

Here, we may proceed as follows:

$$w^k \leftarrow \underset{w \in \mathbb{R}^p}{\arg\min} \, \mathbb{E}_{s,a \sim \mu^k} \left[ \left( \phi_{s,a}^\top w - Q_{s,a}^k \right)^2 \right]$$

$$\theta^{k+1} \leftarrow \theta^k - \eta w^k$$

$$\pi^{k+1} \leftarrow \text{ the log-linear policy defined by } \theta^{k+1}$$

The above can be seen as a special case of $(B)$, by considering the induced updates in state-action space:

$$\widehat{Q}^k = \underset{\widehat{Q} \in \mathcal{F}}{\arg\min}\, \mathbb{E}_{s,a \sim \mu^k}\left[\left(\widehat{Q}_{s,a} - Q_{s,a}^k\right)^2\right] = (s,a) \mapsto \phi_{s,a}^\top w^k$$

$$f_s^{k+1} = \nabla R(\pi_s^k) - \eta \widehat{Q}_s^k$$

$$= \log(e^{\phi_s^\top \theta^k}) - \log(Z_s^k)\mathbf{1} - \eta \widehat{Q}_s^k$$

$$= \phi_s^\top \theta^k - \eta \widehat{Q}_s^k - \log(Z_s^k)\mathbf{1}$$

$$\pi^{k+1} \leftarrow \text{ the policy defined by } \pi_s^{k+1} = P_R(f_s^{k+1}) = \frac{e^{\phi_s^\top \theta^{k+1}}}{\sum_a e^{\phi_{s,a}^\top \theta^{k+1}}}.$$

Note that in the above,

$$f_{s,a}^{k+1} = \phi_{s,a}^\top \theta^k - \eta \phi_{s,a}^\top w^k - \log(Z_s^k),$$

and that $Z_s^k$ is the same for all actions in $s$, hence makes no difference after the projection step..

