# OpenReview forum: "Convergence of Policy Mirror Descent Beyond Compatible Function Approximation"
_ICML.cc/2025/Conference — ICML 2025 poster_

### Official Review · Reviewer_9aB9 · 2025-03-03

**Overall Recommendation:** 4

**Summary:**

This paper establishes an upper bound on the convergence rate of the on-policy policy mirror descent (PMD) algorithm in an agnostic learning setting for discounted MDPs. The authors replace the commonly assumed closure condition with a weaker variational gradient dominance (VGD) assumption. Under this condition, they prove that PMD achieves a convergence rate of  $O(K^{-2/3})$￼with Euclidean regularization and $O(K^{-2/7})$ with negative entropy regularization, where $K$ is the number of iterations. Their analysis frames PMD as a non-Euclidean proximal point method, leveraging a local smoothness property of the value function defined by local norms induced by the policy’s occupancy measure.

## update after rebuttal
I have reviewed the authors’ rebuttal and found their responses satisfactory. My assessment remains unchanged, and I continue to recommend acceptance.

**Claims And Evidence:**

The claims are well-supported by rigorous theoretical proofs and an illustrative example.

**Essential References Not Discussed:**

The discussion of related prior work is thorough.

**Experimental Designs Or Analyses:**

N/A

**Methods And Evaluation Criteria:**

The authors carefully discuss and compare their results with prior work.

**Other Comments Or Suggestions:**

- Empirical validation of the theoretical results would strengthen the paper. Implementing PMD under the VGD condition and comparing its empirical convergence with the derived theoretical bounds could provide valuable insights.
- A careful comparison between the convergence rates under the VGD condition and those under the closure condition would help contextualize the benefits and limitations of the proposed framework. This could clarify whether VGD provides comparable or significantly different guarantees relative to more restrictive assumptions.

**Other Strengths And Weaknesses:**

Overall, this paper is well-structured and provides a solid theoretical contribution to the convergence of PMD under the agnostic learning setup for discounted MDPs.

Strengths:
- Introducing the VGD condition and proving the convergence of PMD without closure conditions are novel contributions.
- Using local smoothness of the value function to establish convergence rate independent of the cardinality of state space is a novel technical contribution.

Weaknesses:
- The paper lacks a comparison between the convergence rate obtained under the VGD condition with existing convergence results under more restrictive conditions, such as the complete policy class or the closure condition.
- There is a lack of experiments to validate the theory, though this is common in RL theory papers.

**Questions For Authors:**

How do the derived rates under VGD (e.g., $O(K^{-2/3})$) compare to known rates under closure conditions (e.g., does VGD degrade rates?)?

**Relation To Broader Scientific Literature:**

N/A

**Theoretical Claims:**

I did not find any issues in the theoretical claims, though I did not verify the deferred proofs in the appendix.

A key component of the analysis is Lemma 2, which establishes the local smoothness of the value function with respect to the local norm induced by the policy’s occupancy measure. The proof relies on value difference lemmas and occupancy measure properties. This local smoothness condition enables the authors to derive convergence rates independent of the state space size.

Another crucial component is the reduction of the PMD setup to an optimization framework for constrained non-convex optimization of locally smooth objectives. The theoretical framework introduced is novel.

---

> ### Author Rebuttal · Authors · 2025-03-31
>
> Thank you for reviewing our work and for your thoughtful comments.
>
> **“Overall, this paper is well-structured and provides a solid theoretical contribution...”** We were happy to read that the reviewer appreciates the value of our contribution, and in particular that the reviewer recognizes **“Using local smoothness of the value function to establish convergence rate independent of the cardinality of state space is a novel technical contribution.”** - we completely agree!
>
> ## Weaknesses
> **“The paper lacks a comparison … the closure condition.” / “A careful comparison between … more restrictive assumptions.”** This is a good point which we will address in our revision. Thanks to your comment, and questions from other reviewers, we have extended the “perfect closure implies VGD” from our Appendix A.2 to hold generally; see below “Relation of VGD and Closure”. Also below, we have included a summary comparing assumptions and rates of ours and prior works “Comparison with prior works”.
>
> **“Empirical validation … could provide valuable insights.”** Our contribution is primarily theoretical and empirical evaluations are outside the scope of our work. We agree that such research may be valuable and should be considered for future work.
>
>
> ## Questions For Authors:
> **“How do the derived rates … degrade rates?)?”** With constant step sizes, the best rate achievable (given closure or in the tabular, exact, complete class setup) is $O(1/K)$. With non constant geometrically increasing step sizes, linear rates (i.e., $O(e^{-cK})$ for some constant $c$) are attainable (again, given closure or a tabular setup). We do not expect these rates to be attainable assuming only VGD. We include additional remarks in the comparison below.
>
> # Relation of VGD and Closure
> We adopt the closure conditions as formalized in Alfano et al. 2023. Let $C_v$ be concentrability coefficient (A2 in Alfano et al.) and $\nu_\star$ the distribution mismatch coefficient (A3 in Alfano et al.).
>
> Then the following relation holds: $\epsilon$-Approximate Closure (A1 in Alfano et al.) $\implies$ $(\nu_\star,H\nu_\star\sqrt{C_v \epsilon})-$VGD.
> Importantly, the $\epsilon_{vgd}=H\nu_\star\sqrt{C_v \epsilon}$ is **exactly** the error floor exhibited in the results of Alfano et al. 2023 (as well as in other works based on closure).
>
> # Comparison with prior works
> The table below summarizes representative papers and their rates for constant step size PMD. Columns state assumptions made in different works.
>
>
>
>
> | Paper                | VGD | Closure Assumptions | Parametric Assumptions | Realizability | Rate        |
> | -------------------- | --- | ------------------- | ---------------------- | ------------- | ----------- |
> | Lan 2023 / Xiao 2022 | Yes | Yes (perfect)       | Yes (Tabular)          | Yes           | $1/K$       |
> | Yuan et al. 2022     | Yes | Yes (approx)        | Yes (Log-linear)       | Yes           | $1/K$       |
> | Ju & Lan 2022        | Yes | Yes (approx)        | Yes (EMaP$^{[a]}$)     | Yes           | $1/\sqrt K$ |
> | Alfano et al. 2023   | Yes | Yes (approx)        | Yes (EMaP$^{[a]}$)     | Yes           | $1/K$       |
> | **Our work**         | Yes | No                  | No                     | No            | $1/K^{2/3}$ |
> |                      |     |                     |                        |               |             |
> - $[a]$ Stands for Exact Mirror and Project
>
>
> ### Remarks
>
> - *Our rate.* We report our $K^{2/3}$ rate for the Euclidean regularization case. More generally, our rates depend on smoothness of the action regularizer, which leads to worse rates for regularizers such as the negative-entropy. We conjecture both that the base rate can be improved to $1/K$ and that dependence on regularizer smoothness can be lifted; however, this seems to require new ideas and we leave it for future work.
>
> - *Non constant step size results.*  With non constant geometrically increasing step sizes, linear rates (i.e., $O(e^{-cK})$ for some constant $c$) are attainable (e.g., Alfano et al. 2023, Xiao 2022, Jhonson et al. 2023). We do not expect this is possible assuming only VGD. The reason these rates are attainable subject to closure is that the algorithm dynamics mimic those of the tabular setting, where policy iteration converges at a linear rate. Assuming only VGD, policy iteration no longer converges (at all), as the policy class loses the favorable structure allowing for convergence of such an aggressive algorithm. This should highlight the value in studying the function approximation setup without closure.
>
> - *Convexity of the policy class.* In our submission we point out the convexity of the policy class as a potential differentiator between our approach and that of Alfano et al., 2023. Following the reviewers comments, we realized this is not the case, and that convexity requirements are similar in our work and that of Alfano et al. 2023. Given closure, our analysis does not require convexity, same as Alfano et al.

---

> > ### Comment · Reviewer_9aB9 · 2025-04-04
> >
> > I appreciate the authors for their rebuttal. Most of my concerns have been addressed, and I am satisfied with the clarifications provided. I will maintain my positive evaluation and recommend acceptance.

---

### Official Review · Reviewer_ch65 · 2025-03-16

**Overall Recommendation:** 3

**Summary:**

The paper addresses the convergence of policy mirror descent method for generalized policy classes ie policy classes with a general parametrization. Most of the prior literature considers either the tabular case where each state can be mapped to any probability vector over the action space, or they consider policy classes which are closed under the PMD update. The main assumption the authors rely on is the variational gradient dominance (VGD) (a PL inequality of sorts for the policy class) of the value function and the convexity of the policy class. The authors argue that the prior assumption over the policy class being closed is stronger than VGD since a closed policy class implies VGD whereas the converse isn't true, making VGD a weaker assumption.

**Claims And Evidence:**

I haven't read through the appendix in detail but the paper in its current form cannot be assessed without a detailed reading of the appendix.
However I do some comments/concerns that I have elaborated on later.

**Essential References Not Discussed:**

I am not well versed in the literature pertaining to the non tabular PMD methods, hence I am unsure of all the important references being included.

**Experimental Designs Or Analyses:**

NA

**Methods And Evaluation Criteria:**

NA

**Other Comments Or Suggestions:**

NA

**Other Strengths And Weaknesses:**

Strengths:
1. Addresses an important problem in the literature
2. Provides bounds that are independent of the cardinality of the state space
3. Relaxes an assumption (from closed policy class to VGD) typically involved in the analysis of these problems

Weaknesses:
1. The paper can use better presentation. In its current form it is hard to delineate what the different assumptions are in different prior works. Also the example in Section 1.2 needs better elaboration. It is not clear at all as to why the policy class here is not closed.
2. In appendix C.3 Proof of theorem 1, line 1034, in the suboptimality expression of $V$, the exploration probability $\epsilon_{expl}$ appears in the denominator. Typically inducing an $\epsilon$ exploration in your policy class leads to a suboptimality which grows $O(\epsilon)$, but here greater the exploration probability smaller the suboptimality, which is contradicting to whats typically observed. I don't understand the reason for this. Also, is there a reason for setting $\epsilon_{expl} = K^{-2/3}$? Cause this seems to lead to a regret that eventually increases in terms of the approximation errors (in theorem 1).

**Questions For Authors:**

What does a closed policy class implying VGD mean in terms of $\epsilon_{vgd}$? Since $\epsilon_{vgd} = \frac{1}{1-\gamma}$ is trivially true for all policy classes, do closed policy classes yield an $\epsilon_{vgd} < \frac{1}{1-\gamma}$?

**Relation To Broader Scientific Literature:**

Convergence of PMD methods for generalized policy classes is indeed an important question that needs to be answered. Especially in the context of Deep reinforcement learning, where the policies are parametrized in terms of the weights of the NN, the theoretical performance of these algorithms is not very well understood.

**Theoretical Claims:**

I haven't examined the proofs in detail.

---

> ### Author Rebuttal · Authors · 2025-03-31
>
> Thank you for reviewing our paper and providing thoughtful comments. We were glad to hear you appreciate our work, in particular, that the reviewer recognizes our paper **“addresses an important problem in the literature”**, and **“relaxes an assumption (from closed policy class to VGD) typically involved in the analysis of these problems”**.
>
>
> # Weaknesses
>
> **1.** Following your comment and those of the other reviewers, we have tightened our comparison with prior works.
>
> ### Comparison of assumptions and rates
> The table below summarizes representative papers and their rates for constant step size PMD. Columns state assumptions made in different works.
>
> | Paper                | VGD | Closure Assumptions | Parametric Assumptions | Realizability | Rate        |
> | -------------------- | --- | ------------------- | ---------------------- | ------------- | ----------- |
> | Lan 2023 / Xiao 2022 | Yes | Yes (perfect)       | Yes (Tabular)          | Yes           | $1/K$       |
> | Yuan et al. 2022     | Yes | Yes (approx)        | Yes (Log-linear)       | Yes           | $1/K$       |
> | Ju & Lan 2022        | Yes | Yes (approx)        | Yes (EMaP$^{[a]}$)     | Yes           | $1/\sqrt K$ |
> | Alfano et al. 2023   | Yes | Yes (approx)        | Yes (EMaP$^{[a]}$)     | Yes           | $1/K$       |
> | **Our work**         | Yes | No                  | No                     | No            | $1/K^{2/3}$ |
> |                      |     |                     |                        |               |             |
> - $[a]$ Stands for Exact Mirror and Project
>
>
> For additional remarks, we refer to our response to Reviewer 9aB9 (“Comparison with prior works”). Regarding the formal relation between VGD and closure, see below “Relation of VGD and Closure”.
>
> ### The example in Section 2.1
> The technical argument is given with full detail in Appendix E.2, where we demonstrate a strong form of closure w.r.t. the optimal policy occupancy (bias-error) does not hold. The simplest way to see that the standard approximation-error closure does not hold globally is since the setup in question is not approximately realizable (we have $V^\star=0$ while $V^\star(\Pi)=H/2$).
>
>
> The goal with this example was not only to show that closure does not hold, but also that the bounds of prior works become vacuous, and for that, additional technical arguments are required. Following your comment, we will revise the exposition of this example, and try to find a better balance between conveying the idea while keeping technical details to the minimum.
>
>
>
> **2.** The $O(\epsilon)$ term you are missing is there through the additive $\delta$ term. We define it in the first line of the proof, line 1013, $\delta=\epsilon_{\rm vgd} + 12\epsilon_{\rm expl}C_\star H^2 A^2$. We suppose it is now clear that the reason for setting $\epsilon_{expl}=K^{-2/3}$ is to balance the two terms. We will highlight the definition of $\delta$ in a display equation so that it cannot be easily missed.
>
> ## Questions for Authors:
> This is a good question, as we show in the paper in Appendix A.2, perfect closure with negative entropy regularization implies $\epsilon_{\rm vgd}=0$. Thanks to your comments and those of the other reviewers, we have added a simple extension of this implication to hold for the general closure conditions as formalized in Alfano et al. 2023.
>
> # Relation of VGD and Closure
> We adopt the closure conditions as formalized in Alfano et al. 2023. Let $C_v$ be concentrability coefficient (A2 in Alfano et al.) and $\nu_\star$ the distribution mismatch coefficient (A3 in Alfano et al.).
>
> Then the following relation holds: $\epsilon$-Approximate Closure (A1 in Alfano et al.) $\implies$ $(\nu_\star,H\nu_\star\sqrt{C_v \epsilon})-$VGD.
> Importantly, the $\epsilon_{vgd}=H\nu_\star\sqrt{C_v \epsilon}$ is **exactly** the error floor exhibited in the results of Alfano et al. 2023 (as well as in other works based on closure).

---

> > ### Comment · Reviewer_ch65 · 2025-04-05
> >
> > Thank you for addressing my questions and comments.
> >
> > I am not incredibly well versed with this domain to assess the contributions with a high confidence, however I am increasing my score since my questions have been addressed.

---

### Official Review · Reviewer_zqsi · 2025-03-17

**Overall Recommendation:** 3

**Summary:**

This paper studies the convergence of policy mirror descent under the variational gradient dominance condition.

**Claims And Evidence:**

Under the variational gradient dominance condition, this work obtains the first convergence rate of PMD. I have some major concerns:
1. I am confused about the claim in Section 1.2 that VGD is not comparable with the other conditions in the literature. Since this is a relaxation of the usual PL-inequality, at least one can compare the theoretical results in this paper with the prior works assuming PL or strong convexity. Following this logic, I suggest the authors providing a table showing how condition (2) is satisfied in the literature and list the corresponding convergence rate of the algorithm. At least, the comparison to [1, 2] should be presented.
2. I am concerned about the assumptions in Lemma 4. Could you show how $D$, $M$ and $\beta$ depend on the parameters of a MDP?
3. I am also concerned about Assumption 1 & 2. Usually in the literature, we need a subroutine to have small $\varepsilon_{\rm act}$ and $\varepsilon_{\rm crit}$. How do you achieve this in your setup?
4. Seemingly Theorem 1 is only a consequence of Theorem 3. Then what is the contribution of this paper?


[1] Lan, G. (2023). Policy mirror descent for reinforcement learning: Linear convergence, new sampling complexity, and generalized problem classes. Mathematical programming, 198(1), 1059-1106.

[2] Zhan, W., Cen, S., Huang, B., Chen, Y., Lee, J. D., & Chi, Y. (2023). Policy mirror descent for regularized reinforcement learning: A generalized framework with linear convergence. SIAM Journal on Optimization, 33(2), 1061-1091.

**Essential References Not Discussed:**

NA

**Experimental Designs Or Analyses:**

NA

**Methods And Evaluation Criteria:**

NA

**Other Comments Or Suggestions:**

NA

**Other Strengths And Weaknesses:**

The presentation of this work should be significantly improved.
1. Introduce notations, assumptions and definitions only when you need them. Move any notions that are not directly relevant to your analysis to the appendix.
2. Avoid redundancy. Many results and notations are introduced twice, first time in Section 1.1 and then again in Section 2. Be precise and consistent.

**Questions For Authors:**

NA

**Relation To Broader Scientific Literature:**

NA

**Theoretical Claims:**

NA

---

> ### Author Rebuttal · Authors · 2025-03-31
>
> Thank you for your thoughtful review and comments.
>
> **1.** We show in the paper (Appendix A.2) that perfect closure in the negative entropy case implies VGD. Following your comment and those of the other reviewers, we have extended this implication to hold for general approximate closure assumptions as formalized in Alfano et al., 2023. A summary comparing our assumptions / bounds with prior works is given below. Columns state assumptions required by the different works. Rates are for constant step size PMD.
>
> | Paper                | VGD | Closure  Assumptions     | Parametric Assumptions | Realizability | Rate        |
> | -------------------- | --- | ------------- | ---------------------- | ------------- | ----------- |
> | Lan 2023 / Xiao 2022 | Yes | Yes (perfect) | Yes (Tabular)          | Yes           | $1/K$       |
> | Yuan et al. 2022     | Yes | Yes (approx)  | Yes (Log-linear)       | Yes           | $1/K$       |
> | Ju & Lan 2022        | Yes | Yes (approx)  | Yes (EMaP$^{[a]}$)     | Yes           | $1/\sqrt K$ |
> | Alfano et al. 2023   | Yes | Yes (approx)  | Yes (EMaP$^{[a]}$)     | Yes           | $1/K$       |
> | **Our work**         | Yes | No            | No                     | No            | $1/K^{2/3}$ |
> |                      |     |               |                        |               |             |
> - $[a]$ Stands for Exact Mirror and Project
>
> We refer to our response to Reviewer 9aB9 (Under “Relation of VGD and Closure” and “Comparison with prior works”) for additional details regarding the comparison. The other paper [2] Zhan et al. 2023 you have mentioned studies the regularized MDP setup, and requires the same set of assumptions as Lan 2023/ Xiao 2022. Note that while VGD is indeed a relaxation of the PL inequality, prior works do not make the PL-inequality assumption directly.
>
> **2.** Lemma 4 is just a building block in the proof of Theorem 1. The connection between the MDP parameters and the parameters stated in the Lemma is given in the proof of Theorem 1; the proof sketch for the Euclidean case is given in the main text in the final paragraph of Section 3.3, and the detailed proof for both Euclidean and negative entropy cases is given in Appendix C.3.
>
> **3.** These are standard assumptions in the general function approximation setup; we expect these to be achieved by optimizing neural network models (e.g., actor and critic networks). Since we do not make any structural assumptions, the corresponding optimization problems are non convex in the network parameter space and thus do not admit provably efficient optimization procedures. See e.g., Alfano et al. 2023 and Xiong et al., 2024 where similar assumptions are made. The assumption on optimality conditions is implied by approximate optimality in function values.
>
> **4.** We are not sure we understand the question, the main contribution of the paper is Theorem 1. In order to prove Theorem 1, we break it into several building blocks. The central building block on the optimization side is Theorem 3. In order to obtain Theorem 1 from Theorem 3, we need to combine it with Lemma 2 (which establishes the key local smoothness property of the value function), and Lemma 4, which links Theorem 3 with the MDP setting.
>
> ## Other Strengths and Weaknesses
> Thank you for your comments, we will revise our exposition for the final version.
>
> Please note however that it is a common practice to provide an overview of the results with only minimal preliminaries as we do in Section 1.1. This is beneficial for the many readers that only want to understand the main results at a high level. That said, if you have any specific suggestions we will be happy to hear and consider them during our revision.

---

### Decision · Program_Chairs · 2025-05-01

**Decision:**

Accept (poster)

**Comment:**

This paper analyzes policy mirror descent (PMD) for general policy classes (where policy closure conditions are not necessarily satisfied), and obtains upper bounds on the convergence rate to the best (in class) policy.

The reviewers unanimously agree that the paper is well-written, and its contributions merit acceptance. After carefully reading the rebuttal/discussion, I tend to agree. Please incorporate the following comments in the final version of the paper. In particular, addressing the following concerns will help strengthen the current version of the paper:
- Improve the presentation and include the table (in response to all the reviewers) in the main paper.
- Include an explanation as to when/how assumptions 1 and 2 can be satisfied in simple settings (e.g., linear function approximation)
- Better explain the example in Section 2.1 (response to Rev. ch65)
- Include the comments about the rates obtained assuming closure vs those obtained using VGD (response to Rev. 9aB9)